# Design of an Autonomous, Sustainable Sharing Mobility Solution Aimed to Mobility-Disabled Individuals

**Leonardo Frizziero** , **Giulio Galiè** * , **Martina Aldrovandi, Silvia Franco and Elisa Rana**

Department of Industrial Engineering (DIN), Alma Mater Studiorum—Università di Bologna,
40136 Bologna, Italy
* Correspondence: giulio.galie2@unibo.it

**Abstract:** Recent analysis has shown deteriorating traffic conditions in urban areas, caused by an increase in the motorization rate, which has risen to 66.6 vehicles per 100 inhabitants. As a result of the pandemic, individuality has grown, hence private vehicles are becoming more prevalent whilst public transport and sharing are negatively affected. Therefore, European policies have encouraged and innovated more sustainable mobility. Thus, the developed project aims to achieve more efficient mobility and more sustainable environments, towards social and economic well-being. The proposed means of transport aims to appeal to an audience with a reduced ability to drive a car as intended. The IDeS methodology was applied to develop a self-driving, urban micro mobility vehicle, aimed to give enough room and equipment for people with moving disabilities. The innovation of the IDeS method is state-of-the-art and ought to satisfy current product needs, which leads to an innovative micromobility vehicle and portrays a design for a car that will help to close the gaps in urban mobility. These design processes, which are distinguished by the fusion of several industrial techniques, enabled the development of a plan that addresses current mobility issues for disabled people and opens to new mobility prospects.

**Keywords:** industrial design structure (IDeS); quality function deployment (QFD); stylistic design engineering (SDE); sharing mobility; design for all

## 1. Introduction

The recent pandemic period has brought to light the changing demands and needs of people to travel in urban centres and provided some interesting insights into the decisions and changes that can be undertaken in the near future. In particular, sharing services, where simultaneous or sequential sharing by several users for the same vehicle seems to be the key to the service. Therefore, the opportunities and limitations of sharing vehicles have been analyzed and put under the spotlight [1]. It was possible to analyze how the car fleet continued to grow in 2020 despite the economic crisis, and cars on the road are just under 40 million (39.7, 0.4 percent more than in 2019) with the motorization rate rising to 66.6 vehicles per 100 inhabitants. As a counterpoint to this trend, it is possible to imagine a parallel and alternative model to partially replace the travel needs covered today by private vehicles, which demonstrates an often poor overall efficiency between the space occupied, and therefore the traffic generated, and the number of people moved, since the occupants on board result statistically with an average value of less than half of the total seats available [2]. To find a solution to these problems [3], a revolutionary future scenario on urban mobility in 2060 was imagined. It is assumed that many people will give up owning a vehicle and choose alternative means of transport, therefore car manufacturers will push towards the production of sharing or self-driving vehicles. Governments will have to give incentives for the creation of new dedicated infrastructure. Furthermore, it is supposed that numerous traffic-restricted zones will be introduced to preserve the livability of the urban environment and reduce pollution in cities, especially in areas characterised

by small spaces and high population density, such as historic city centers. As a result, cities will be less accessible [3,4]. This change certainly turns out to be of great magnitude, and the most effective way to make it happen, and thus entice citizens to replace their vehicles with shared alternatives, is for these services to be very functional, efficient, and require minimal user effort. With these considerations in mind, the concept for a self-driving sharing vehicle was devised that would meet the needs of the broad target audience using concepts derived from the design for all philosophy, and would make a high level of ubiquity attainable, with extremely simple operation, and allowing it to be used even by people without a driver's license [5].

What makes an area sustainable is precisely the level of accessibility it can offer to citizens and all those who need to reach the city. Accessibility for the frail means improving the quality of life overall by reducing stressors. Indeed, by best employing technological development we could avoid the risk of the latent conflict between green and inclusion, which risks making environmentally sustainable choices non-inclusive. Developments in technology and their application will enable social change by making freedom of movement untethered from the use of private vehicles as we know them today, moving toward connected and smart roads traversed by self-driving vehicles [3].

The application of the IDeS method, which covers all industrial design phases, allows the aesthetic and technical completeness of the vehicle concept: project setup, product development stage, and ending with the production startup [6]. This optimizes and speeds up time-to-market, as a well-defined product is obtained [7]. A division of the article into sections was performed following all the stages of the IDeS methodology (Figure 1) [6]. The design approach starts from the environmental analysis, which is useful to define the target area in which it operates, followed by the whole analytical part of the method. In order to deliver unique insights by offering fresh and efficient solutions, the creative means will need to address all the needs identified based on the discovered limits. It will be crucial to explore potential scenarios while considering how society will surely change and what wants and requirements there would be. This work aims to design, using the IDES methodology, an inclusive and accessible self-driving sharing vehicle for people with disabilities, in which they can travel alone or accompanied by another person. This smart and sustainable vehicle is designed for 2040 in order to change the paradigm of micro-mobility of disabled and non-disabled persons.

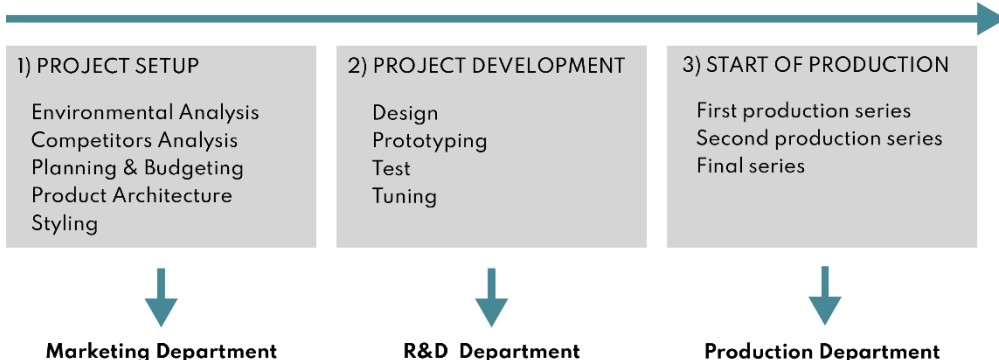

**Figure 1.** Industrial Design Structure (IDeS) outline.

## 2. Materials and Methods

The project was created using the IDeS approach, which guides the company through the product development process by merging the technical and stylistic aspects of product design, from which stylistic design engineering (SDE) is generated. This method, introduced by Lorenzo Ramacciotti, ex-CEO of the design house Pininfarina, reveals a systematic methodology to make an automaker's style more consistent and recognizable worldwide. IDeS combines tools for technical analysis and comparison of existing products on the mar-

ket, such as quality function deployment (QFD), benchmarking, and top-flop analysis [6]. This strategy allows for the management of industrial design phases by involving the business organisation and sharing resources directly. In order to produce a high-quality product that is competitive in the market, it was discovered via the analysis of various case studies created using the IDeS technique that these products are customer-centric but also consider industrialization, technological, and financial restrictions. Project setup, product development, and production startup are the three key steps that make up IDeS. The designer is mainly concerned with the first two phases in which the product is conceived and takes shape throughout its design [6]. The figure summarises and collects all the stages of the IDeS process (Figure 2).

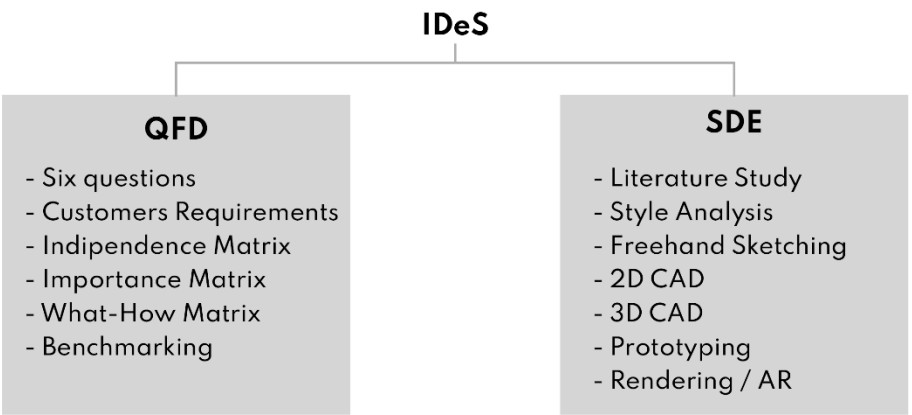

**Figure 2.** Stages of the industrial design structure.

The IDeS method makes it possible to make a thorough study of what is already on the market and the best features of competitors' products, identifying the degree of innovation to be achieved and overcome. From the research carried out on the described method, it has been shown to be effective for the design of a product intended for a specific market and to be put into production in a faster time. Finally, through this method, an attempt is made to meet and satisfy the needs of the market, customer and industry.

Moreover, the project setup stage is composed of the following phases: (a) environment analysis, (b) market analysis, (c) budgeting and planning, (d) architecture definition and (e) styling. Product development is composed of these phases: (a) 3D modelling (CAD), (b) prototyping, (c) testing and (d) optimization and final tuning (Figure 3).

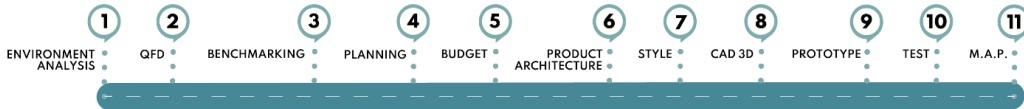

**Figure 3.** Project steps (road map).

### 2.1. Environment Analysis

In this stage, one learns about the environment and the target market in which the product to be designed will fit. Understanding the manufacturing, commercial, and cultural environments is crucial.

### 2.1.1. Quality Function Deployment (QFD)

Quality function deployment (QFD) is a structured approach to defining customer needs or requirements and translating them into specific plans to produce products to meet those needs [6]. Customer needs are then transformed into internal design standards for the firm throughout the product development process. These criteria are often general product qualities (measurable quantities) that will help to satisfy the needs of the client [8]. This form of the survey may be used in a wide range of research topics, which is why

more and more businesses are including it into their market analyses. The "wh-questions" (who, what, how, where, when, and why) and the interaction matrices make up the two components of the QFD (of relative importance and that of dependence-independence). In order to further describe the project's insertion overview, the "wh-questions" must be answered in order to provide a list of qualitative needs from the customer's perspective (target, functionality, location, why it was done and what issues it solves). The relative relevance matrix compares these needs to one another. They are interpolated from each other by assigning values from 0 to 2. Zero is used when the requirement on the row is more important than the column, and 1 is used when they are considered of the same importance, and 2 is used if the requirement on the column is more important than the one on the row. Doing the sum of the values of each column gives the ranking in order of importance, and the columns that get a higher figure are those to be considered as design features. In the dependence-independence matrix, the requirements are compared to define how dependent they are on each other. The values 0, 1, 3, 9 are assigned. Zero is used if the row requirement is completely independent of the column requirement, 1 if it is not very dependent, 3 if it is very dependent, and 9 if it is totally dependent. Adding up the values of each column gives a ranking, and the requirements with the highest value are the most independent and will be part of a list of characteristics to be given more consideration [9].

QFD was developed to solve three general problems in the Western industry: loss of information during the product development cycle, inattention to customer needs and the different interpretations of specifications by the various industrial departments involved. This is also demonstrated by several papers citing its use [6].

### 2.1.2. Benchmarking and Top-Flop Analysis

A technique for evaluating the products of rival companies on the market is competitor analysis (also known as benchmarking). This phase's goal is to determine what technological traits the proposed product must possess in order to stand out from the competition. Benchmarking consists of a quantitative analysis, specifically comparing (using a matrix) the technical characteristics of competitor products. The various features have been colored for easier graphical feedback. The best data for each characteristic are highlighted in green, and the worst in red, which are then reported in the Top-Flop analysis at the bottom of the table. Then Top-Flop analysis is performed by selecting for each technical specification the best (top) and worst (flop) performance. From the difference of the tops and flops, the degree of innovation that the product will need to have in order to be innovative is inferred. Next, the what-what matrix is created where the best technical features identified in the benchmarking are compared with the most important requirements from a customer perspective (extrapolated from the interrelationship matrices). The matrix is structured with the columns containing the technical specifications and the rows with the requirements in customer perspective. Values 0, 2, 4, 6, 8, and 10 are assigned based on how much the technical specifications influence the customer requirements. Summing the values of each column yields a ranking of the features to be innovated and that will drive the design course [10].

### 2.2. Stylistic Design Engineering (SDE)

SDE is a widely used industry process for a new car design. It is a method that involves sequential processes that, when successfully completed, may result in designs that are in keeping with both the history of the brand for which they are being created and the current market trends. Studying fashion trends serves as the first step in the process [6]. In the case where the vehicle is intended for a specific company, the history of the company, the key moments in its history and its stylistic past must be analyzed. After the study is the paper drawing stage. Thanks to the knowledge and shapes acquired in the previous stage, it is now easy to imagine and trace shapes that are as consistent as possible with the target company and its style. In this way, the designer's personal style and stroke are preserved and emphasized, while remaining within solid guidelines that help him or her during the sketching process. Different sketches are then made for each of the style proposals:

natural with softer and more organic shapes, stone with more rigid and squarish shapes, retro with more classic shapes, and advanced with more futuristic and innovative shapes. After selection, the last remaining sketch is edited and refined until it reaches a stylistically consistent and satisfactory level. After the selection of the sketch of greatest interest, it is time to transform it into a rigorous 2D computer drawing. This is a very important step because in freehand sketches the shapes often tend to be very "emotional" and obviously disproportionate. Then 2D drafting comes to the rescue, in which, being very rigorous, suddenly makes the lines much more realistic and allows us to judge the model for what it will be. The next step is the 3D model. At this stage the rendering of the model shapes will not completely match the 2D lines. Once the model is finished, the correct proportions and style concepts born in the sketching phase are immediately visible. Thanks to 3D modeling, it is possible to have a realistic product shape. However, monitor visualization has limitations, such as the correct visualization and definition of surface curvature, and the always very difficult handling of proportions. To achieve this, it is necessary to create a physical model of the design. With a physical model made, it is possible to correctly assess the quality of the surfaces and apply changes to the 3D model in CAD. At this point comes the important stage of communication, which is to convey the consistency, organicity, and beauty of the lines. Digital renderings are used to do this. Thanks to the renderings, it is possible, through the assignment of materials to the various components of the product, to make the three-dimensional models look realistic, exactly as they will be after they are put into production, and thus enabling the virtual recreation of scenarios and places of everyday use of the product (Figure 4).

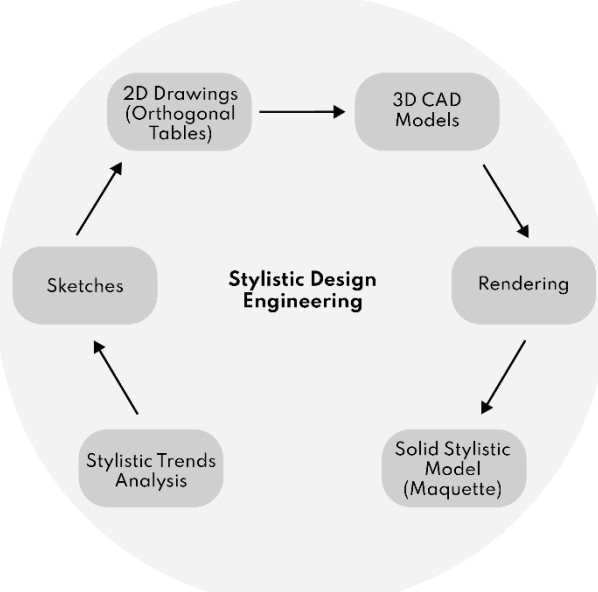

**Figure 4.** Main phases of stylistic design engineering.

*2.3. Design Phase*

2.3.1. Design Engineering

During this phase, the main technical characteristics of the design are determined using a digital model, starting with the dimensions of the overall dimensions and then moving on to the selection of the most suitable technologies and materials for use. Thereafter, the second part of this project phase includes testing and prototyping. This step includes the shape choice, size, and materials that will characterize the product, using 2D and 3D modeling CAD software. Next, the design is developed in more detail, and defining in more detail some important aspects such as usability and component parts.

### 2.3.2. Virtual Prototyping

In the IDeS methodology, prototyping is a very important stage in product development because it allows the understanding of shapes, proportions and surfaces, thanks to the possibility of comparison with a realistic model. The first is a virtual model that is made using 3D modeling CAD software and is parametric as it allows changes to be applied very quickly. The second prototype is achieved by rendering the 3D model, using dedicated software, ensuring a strong aesthetic impact and optimal visual perception of the applied materials. The third prototype is achieved by additive 3D printing technology, with which the object can be physically verified in all its forms. Finally, it is possible to conclude the prototyping phase with a 1:1 scale working model.

### 2.3.3. Testing

The testing phase is used to verify whether the innovation goal that was set, was successfully achieved. By carefully checking the prototypes made, the characteristics of the product are thoroughly examined and the old value of $\Delta$ is compared with the newest value of $\Delta$.

### 2.3.4. Redesign

The IDeS method concludes with the redesign phase, where different revisions can be implemented, taking up the last two phases previously described and aiming for further product improvement.

### 2.4. Planning

Before starting the project, a list of macro-phases was drawn up to be completed through the work breakdown structure (WBS). The time frame corresponds to the actual project duration and is set from October to January. In the Gantt plan (Figure 5), a weekly deadline was estimated for each project phase to check that the workflow is effective. Each team member within the design will have different responsibilities and tasks to take care of personally.

The macro-phases of the WBS are as follows:

1. Environmental analysis
2. Competitors analysis
3. Benchmarking
4. Product architecture
5. Sketch SDE
6. 2D CAD
7. 3D CAD
8. Budget
9. FEM analysis
10. Rendering
11. Prototyping

### 2.5. Budget

The R&D (research and development) budget schedule was divided into three parts: one based on the cost of individuals' work performance, one on the cost of materials aimed at prototype development, and another concerning the equipment needed to make the prototypes (Table 1). The total cost to be incurred was calculated for each table. A budget assumption was made on a 3-year project development perspective. Taking what are the activities described earlier in the Gantt plan and associating each with a certain number of workers with different specializations, the total cost of performance was calculated. Three categories of professional figures were identified: engineers, designers and laborers, who will work 8h per day for a weekly period of 5 working days, with an average salary of 50 euros per hour, 40 euros per hour and 30 euros per hour, respectively.

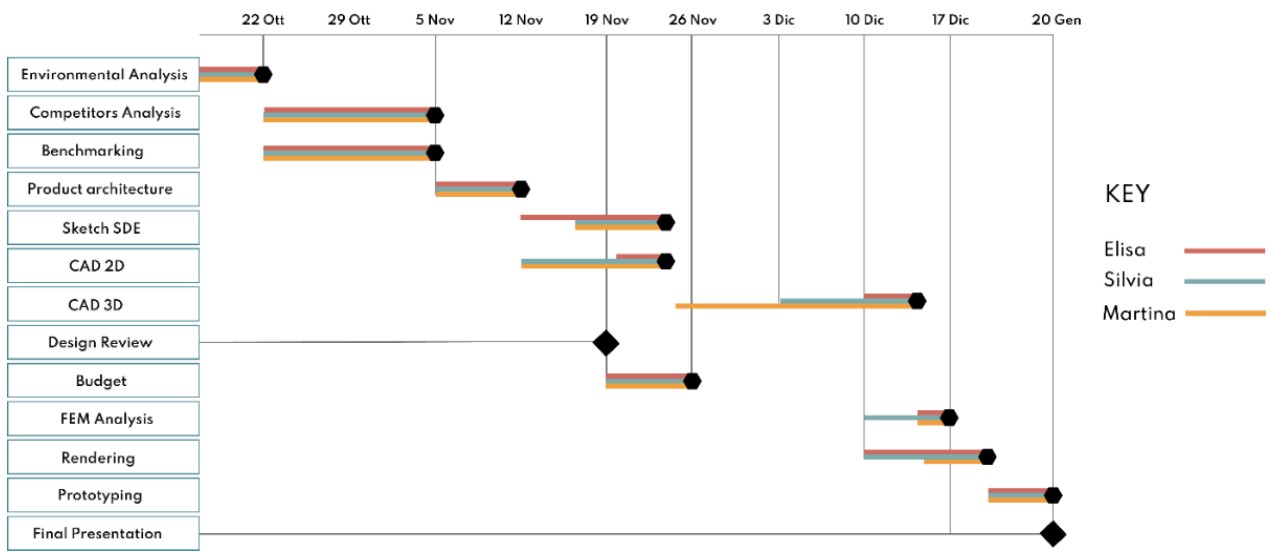

**Figure 5.** Gantt plan.

**Table 1.** R&D budget.

| | PERFORMANCE COSTS | | | | | | | | MATERIAL COSTS | | EQUIPMENT COSTS | |
| --- | --- | --- | --- | --- | --- | --- | --- | --- | --- | --- | --- | --- |
| | Engineer (50 €/h) | | Designer (40 €/h) | | Worker (30 €/h) | | | | | | | |
| Task | No. of Employee | Cost | No. of Employees | Cost | No. of Employees | Cost | N° Hour | Total Cost | Number of Pieces | Cost per Piece | Piece | Cost |
| Environmental analysis | 3 | 150 | | | | | 480 | 72,000 | | | Body | 500,000 |
| Competitors analysis | 3 | 150 | | | | | 480 | 72,000 | | | Frame | 450,000 |
| Product architecture | 5 | 250 | 3 | 120 | | | 960 | 355,200 | | | Windows | 590,000 |
| Styling | 1 | 50 | 4 | 160 | 7 | 210 | 960 | 403,200 | | | External accessories | 100,000 |
| | | | | | | | | | | | Lighting | 98,000 |
| Design | 60 | 3000 | | | | | 1440 | 4,320,000 | | | Interiors | 450,000 |
| Prototyping | 10 | 500 | 1 | 40 | 50 | 1500 | 1440 | 2,937,600 | 5 | 450,000 | Interface and software | 200,000 |
| Test | 1 | 50 | | | 30 | 900 | 1440 | 1,368,000 | _ | 230,000 | Electrical Component | 142,000 |
| Re-design | 60 | 3000 | 5 | 200 | | | 960 | 3,072,000 | | _ | Engine and mechanic | 120,000 |
| Prototyping | 10 | 500 | 5 | 200 | 50 | 1500 | 960 | 2,112,000 | 10 | 480,000 | | |
| Test | 1 | 50 | | | 30 | 900 | 960 | 912,000 | _ | 230,000 | | |
| Deliberation | 2 | 100 | 1 | 40 | 15 | 450 | 960 | 566,400 | | | | |
| | TOTAL | | | | | | | 16,190,400 € | TOTAL | 7,510,000 € | TOTAL | 2,650,000 € |
| | | | | | | | | | | | TOTAL | 26,350,400 € |

### 2.6. Environment Analysis

Designing a smart, sustainable method of urban transportation was the project's original aim. A summary of the pros and cons of micro-mobility sharing transport was made, considering the most relevant problems observed in various articles [11,12], and examining the benefits that encourage people to use this kind of service but more crucially the drawbacks that can serve as a signal for general development (Figure 6).

| PROS | CONS |
|------|------|
| - Freedom<br>- Widespread<br>- No costs for vehicle ownership<br>- Access to restricted traffic areas<br>- Flexibility of use<br>- Reduction of CO2 emissions<br>- Time saving<br>- Integration with other transport vehicles | - Bookable with short notice<br>- Badly parked<br>- Vehicles not usable by disabled people<br>- Risk of not finding the vehicle after the stop<br>- Require dedicated infrastructure |

**Figure 6.** Sharing mobility: advantages and drawbacks.

Since it is a very wide and complicated subject, statistics on mobility trends were examined (Figure 7) in relation to various means of sharing. The graph below shows how car sharing had a dramatic decline in the 2020s and how open and private-use automobiles were favoured [1,13]. This is a negative effect of the pandemic, as sanitation has turned into a crucial and necessary factor in individual transportation selection.

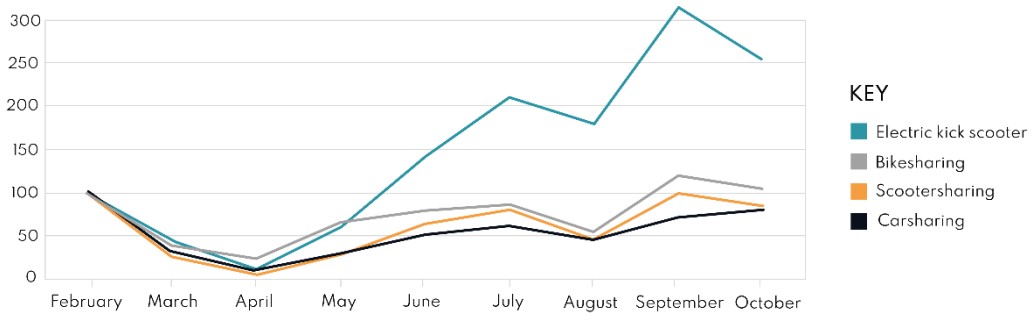

**Figure 7.** Daily rentals per service in 2020. Data collected by the National Sharing Mobility Observatory.

After an in-depth examination of all available means of transport, they are not fit for private and single usage by those with limited mobility. Among all the options evaluated, three were chosen: microcars, e-bikes, and scooters. They were thought to have the best attributes in terms of micro-mobility, but mainly because they are most aligned with the project's goal. The following qualities must be considered: license-free operation, vehicle compactness, ease of use, ubiquity, and access to LTZs [14]. During this additional examination, the issue of the lack of accessible public transportation vehicles for individuals with disabilities emerges prominently. As a result, substantial study was conducted on these people's urban mobility demands and challenges. It can be shown that, according to a Unipolis association forecast based on ISTAT statistics, persons with limited mobility will grow by 25% by 2060 owing to an ageing population (Figure 8) [15]. Since there is a present and future need to adapt urban transportation to a different and more inclusive public, this unequal increase will result in significant change.

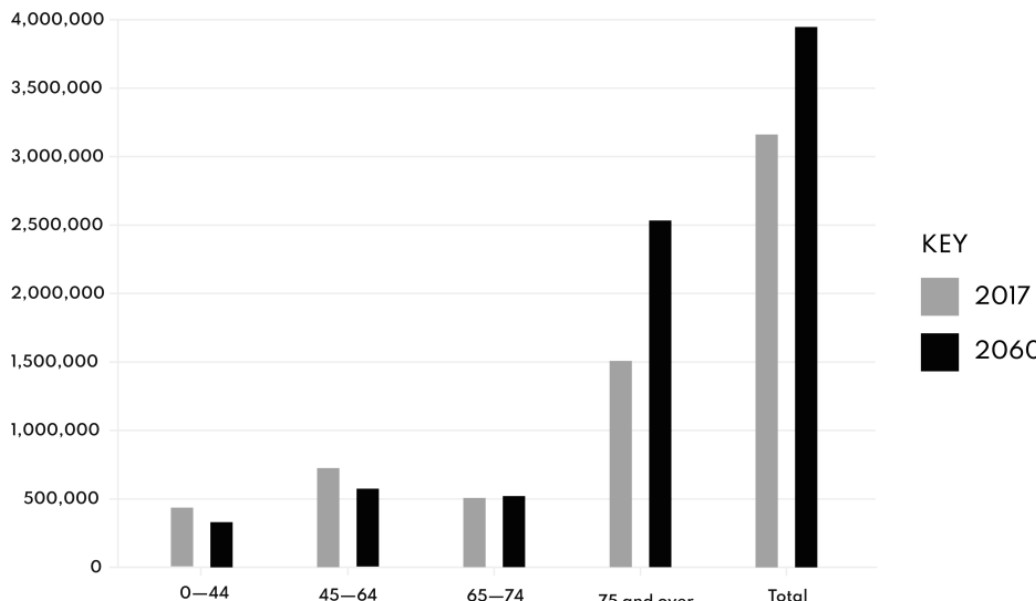

**Figure 8.** Age distribution of the resident population with impairments in Italy through 2017 and an estimate until 2060, developed by the Unipolis foundation using ISTAT data.

ISTAT data was studied to better understand how persons with limited mobility travel in cities on a daily basis (Figure 9) [16]. Every day, 300,000 disabled persons use private automobiles, either as drivers or passengers, to commute to work, while just 7.4 percent use public transportation.

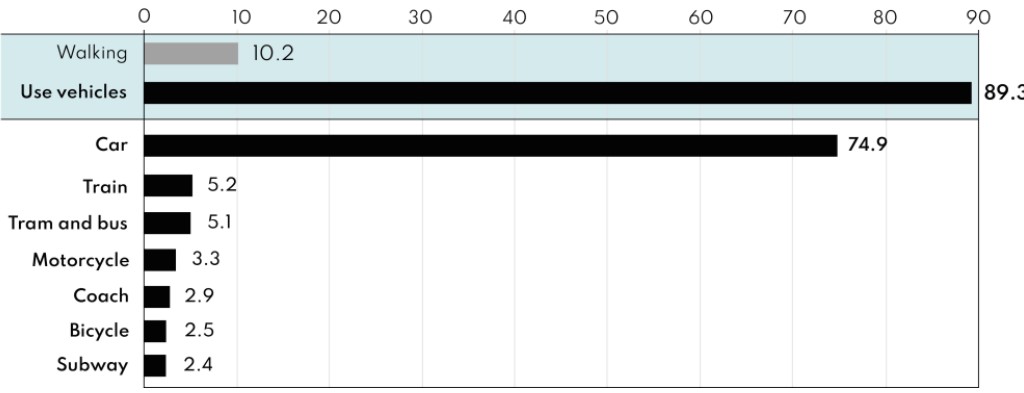

**Figure 9.** Types of vehicles used by people with disabilities.

Due to the low number of persons with disabilities who use public transit, questions have been raised concerning why these people chose private automobiles. Several causes have been identified: the infrastructure is frequently insufficient and becomes inaccessible; there are still cars that do not have ramps; and the interior areas of the vehicles are inadequate owing to capacity or a lack of safety devices [17].

Once the mobility trends were identified, various means of micro-mobility (scooters, segways, hoverboards, electric bicycles, micro-cars, scooters, and electric city cars) were analyzed to determine which means are best suited for urban travel. Based on the technical specifications found on the manufacturers' official websites, we have reinterpreted the data, schematized the features and identified the pros and cons of each type of vehicle in order to get a general overview of the field of use (Table 2).

**Table 2.** Vehicles for urban mobility: pros and cons.

| | Pros | Cons |
|---|---|---|
| **E-BIKE** | No driving license required<br>Easy to use<br>Widespread<br>Avoid the traffic<br>Not much cost<br>Access to restricted traffic areas | Single seat<br>Limited travel distance<br>Low load capacity<br>Does not have coverage<br>Not for all |
| **ELECTRIC KICK SCOOTER** | Compact and quick<br>No driving license required<br>Easy parking<br>Avoid the traffic<br>Widespread<br>Access to restricted traffic areas | Not usable by all<br>Single seat<br>Absence of luggage rack<br>Large number of accidents<br>Does not have coverage |
| **HOVERBOARD** | Compactness<br>Avoid the traffic<br>Low cost<br>Transportability when not used<br>No driving license required | Not usable by all<br>Poor stability<br>Does not have coverage<br>You must learn to use it<br>Absence of luggage rack<br>Usable for short distances |
| **SEGWAY** | Quick<br>Avoid the traffic<br>Sturdy wheels<br>No driving license required<br>No physical effort required<br>Access to restricted traffic areas | Poor stability<br>Not usable by all<br>You must learn to use it<br>Usable for short distances<br>Absence of luggage rack<br>Does not have coverage |
| **SCOOTER** | Quick<br>Avoid the traffic<br>Traveling long distances<br>Can have 2 seats<br>Access to restricted traffic areas<br>It has a luggage rack | Driving license required<br>Not usable by all<br>Large number of accidents<br>Not all have coverage<br>Limited luggage rack capacity |
| **MICROCAR** | Traveling long distances<br>Two seats<br>Protection in case of rain<br>Access to restricted traffic areas<br>Requires little parking space | Does not avoid traffic<br>Need at least the license AM<br>High cost<br>Small luggage compartment<br>Low safety |
| **CITY CAR** | Traveling long distances<br>It has the trunk<br>It has more than two seats<br>Safety<br>Protection in case of rain | Driving license required<br>Does not avoid traffic<br>Cannot go everywhere<br>Not accessible to all<br>High cost<br>Needs space for parking |

*2.7. Quality Function Deployment (Q.F.D)*

In the phase of marketing analysis, we adopted the QFD (quality function deployment) approach to develop the list of design needs for our vehicle by asking six questions.

Who will be the biggest user of our vehicle? People with impairments or limited mobility, kids, adults, and any unlicensed individuals over the age of 16. What is the product's purpose? A small, shared vehicle that may be customized to meet the needs of different persons. How will buyers interact with the product? You may ride standing or sitting, as a passenger or driver, and you can even use walking aids and wheelchairs.

Where will the product be used by customers? City, suburbs, metropolis, and historic districts. When will the product be used? Every day, 24 h a day, for short durations. What is the purpose of the product? It demolishes structural and psychological obstacles. The vehicle's goal is to create communities that are better equipped and adaptable to people's demands, while still being environmentally friendly.

The six questions resulted in the following requirements: (1) travel time, (2) safety, (3) adaptability, (4) compactness, (5) noise, (6) sustainability, (7) affordability, (8) inclusiveness, (9) autonomy, (10) modularity, (11) capacity, (12) comfort, (13) power supply, (14) waiting, (15) ergonomics, (16) practicality, (17) aesthetics, (18) durability, (19) availability, (20) ubiquity, and (21) integrability (Table 3).

**Table 3.** Relative importance relationship matrix.

| | 1 | 2 | 3 | 4 | 5 | 6 | 7 | 8 | 9 | 10 | 11 | 12 | 13 | 14 | 15 | 16 | 17 | 18 | 19 | 20 | 21 |
|---|---|---|---|---|---|---|---|---|---|---|---|---|---|---|---|---|---|---|---|---|---|
| 1. Travel time | 1 | 1 | 1 | 0 | 0 | 0 | 2 | 1 | 2 | 1 | 1 | 1 | 0 | 1 | 0 | 1 | 0 | 1 | 2 | 2 | 1 |
| 2. Safe | 1 | 1 | 1 | 0 | 0 | 0 | 2 | 1 | 2 | 1 | 1 | 1 | 0 | 1 | 0 | 1 | 0 | 1 | 2 | 2 | 1 |
| 3. Adaptable | 1 | 1 | 1 | 0 | 0 | 1 | 1 | 1 | 1 | 0 | 0 | 1 | 1 | 1 | 0 | 1 | 0 | 1 | 1 | 2 | 0 |
| 4. Compact | 2 | 2 | 2 | 1 | 1 | 2 | 2 | 2 | 2 | 1 | 1 | 2 | 2 | 2 | 1 | 2 | 0 | 2 | 2 | 2 | 1 |
| 5. Noisy | 2 | 2 | 2 | 1 | 1 | 2 | 2 | 1 | 2 | 1 | 1 | 1 | 2 | 2 | 1 | 2 | 1 | 2 | 2 | 2 | 2 |
| 6. Sustainable | 2 | 2 | 1 | 0 | 0 | 1 | 1 | 1 | 1 | 0 | 0 | 0 | 1 | 1 | 0 | 1 | 0 | 1 | 1 | 1 | 1 |
| 7. Economical | 0 | 0 | 1 | 0 | 0 | 1 | 1 | 1 | 1 | 0 | 0 | 1 | 1 | 1 | 0 | 1 | 0 | 1 | 1 | 1 | 0 |
| 8. Inclusive | 1 | 1 | 1 | 0 | 1 | 1 | 1 | 1 | 1 | 0 | 0 | 1 | 1 | 1 | 0 | 1 | 0 | 1 | 1 | 1 | 0 |
| 9. Autonomous | 0 | 0 | 1 | 0 | 0 | 1 | 1 | 1 | 1 | 0 | 0 | 1 | 0 | 1 | 0 | 1 | 0 | 1 | 1 | 1 | 1 |
| 10. Modular | 1 | 1 | 2 | 1 | 1 | 2 | 2 | 2 | 2 | 1 | 1 | 1 | 2 | 2 | 1 | 2 | 2 | 2 | 2 | 2 | 1 |
| 11. Spacious | 1 | 1 | 2 | 1 | 1 | 2 | 2 | 2 | 2 | 1 | 1 | 2 | 2 | 2 | 1 | 2 | 0 | 1 | 2 | 2 | 2 |
| 12. Comfortable | 1 | 1 | 1 | 0 | 1 | 2 | 1 | 1 | 1 | 1 | 0 | 1 | 1 | 1 | 0 | 2 | 1 | 2 | 2 | 2 | 1 |
| 13. Power source | 2 | 2 | 1 | 0 | 0 | 1 | 1 | 1 | 2 | 0 | 0 | 1 | 1 | 1 | 0 | 1 | 0 | 1 | 1 | 1 | 1 |
| 14. Waiting time | 1 | 1 | 1 | 0 | 0 | 1 | 1 | 1 | 1 | 0 | 0 | 1 | 1 | 1 | 0 | 1 | 0 | 1 | 1 | 1 | 0 |
| 15. Ergonomic | 2 | 2 | 2 | 1 | 1 | 2 | 2 | 2 | 2 | 1 | 1 | 2 | 2 | 2 | 1 | 2 | 1 | 2 | 2 | 2 | 1 |
| 16. Practical | 1 | 1 | 1 | 0 | 0 | 1 | 1 | 1 | 1 | 0 | 0 | 0 | 1 | 1 | 0 | 1 | 0 | 1 | 1 | 1 | 1 |
| 17. Appearance | 2 | 2 | 2 | 2 | 1 | 2 | 2 | 2 | 2 | 2 | 2 | 1 | 2 | 2 | 1 | 2 | 1 | 2 | 2 | 2 | 2 |
| 18. Resistant | 1 | 1 | 1 | 0 | 0 | 1 | 1 | 1 | 1 | 0 | 1 | 0 | 1 | 1 | 0 | 1 | 0 | 1 | 1 | 1 | 1 |
| 19. Available | 0 | 0 | 1 | 0 | 0 | 1 | 1 | 1 | 1 | 0 | 0 | 0 | 1 | 1 | 0 | 1 | 0 | 1 | 1 | 1 | 0 |
| 20. Widespread | 0 | 0 | 0 | 0 | 0 | 1 | 1 | 1 | 1 | 0 | 0 | 0 | 1 | 1 | 0 | 1 | 0 | 1 | 1 | 1 | 1 |
| 21. Can be integrated | 1 | 1 | 2 | 1 | 0 | 1 | 2 | 2 | 1 | 1 | 0 | 1 | 1 | 2 | 1 | 1 | 0 | 1 | 2 | 1 | 1 |
| TOTAL | 23 | 23 | 27 | 8 | 8 | 26 | 30 | 27 | 30 | 11 | 10 | 19 | 24 | 28 | 7 | 28 | 6 | 27 | 31 | 31 | 19 |

The requirements that scored highest and thus will have the most relevance to the project are availability, ubiquity, affordability, autonomy, and expectation. Next, we entered the same requirements into the dependence/independence matrix (see Table 4). Affordability, capillarity, compactness, flexibility, and availability received the highest scores and so are the most independent.

*2.8. Benchmarking and Top-Flop Analysis*

The competitor's analysis was carried out on the three categories of sharing vehicles considered to be the best for urban mobility: microcars, e-bikes, and scooters. Microcars were selected because despite being very compact, they have two seats and some even have a trunk; these features make it practical and easy to use in the urban environment. The e-bike was selected because of its ease of use and the ability to get almost anywhere and thus be very ubiquitous. The scooter, on the other hand, was selected because it is a vehicle that is rapidly growing in use. All the vehicles that were analyzed have electric power because it is the most compatible with the sharing mode. Benchmarking was divided into three parts (one for each vehicle) and common technical characteristics were chosen in order to compare. For each benchmarking (see Tables 5–7), top-flop analysis was performed (Figures 10–12), and the degree of innovation to be achieved and exceeded was calculated. The benchmarking method is explained more thoroughly in Section 2.1.2.

**Table 4.** Dependence/ Independence Matrix.

| | 1 | 2 | 3 | 4 | 5 | 6 | 7 | 8 | 9 | 10 | 11 | 12 | 13 | 14 | 15 | 16 | 17 | 18 | 19 | 20 | 21 | DEPENDENCE |
|---|---|---|---|---|---|---|---|---|---|---|---|---|---|---|---|---|---|---|---|---|---|---|
| 1. Travel time | x | 1 | 0 | 0 | 0 | 0 | 3 | 0 | 9 | 0 | 3 | 0 | 9 | 9 | 0 | 0 | 0 | 0 | 9 | 3 | 3 | 49 |
| 2. Safe | 1 | x | 3 | 9 | 1 | 3 | 9 | 3 | 0 | 0 | 9 | 3 | 1 | 0 | 3 | 0 | 0 | 9 | 0 | 1 | 1 | 56 |
| 3. Adaptable | 0 | 1 | x | 3 | 0 | 1 | 9 | 9 | 3 | 9 | 9 | 3 | 3 | 0 | 3 | 9 | 0 | 1 | 9 | 9 | 9 | 90 |
| 4. Compact | 1 | 9 | 3 | x | 0 | 1 | 3 | 0 | 3 | 9 | 9 | 3 | 3 | 0 | 3 | 1 | 0 | 1 | 1 | 0 | 3 | 53 |
| 5. Noisy | 0 | 0 | 0 | 1 | x | 0 | 1 | 0 | 0 | 0 | 3 | 1 | 9 | 0 | 0 | 0 | 0 | 0 | 0 | 0 | 0 | 15 |
| 6. Sustainable | 9 | 0 | 3 | 1 | 0 | x | 9 | 0 | 9 | 1 | 3 | 0 | 9 | 0 | 0 | 0 | 0 | 3 | 3 | 1 | 0 | 51 |
| 7. Economical | 9 | 9 | 9 | 3 | 3 | 9 | x | 1 | 3 | 9 | 3 | 9 | 9 | 0 | 3 | 1 | 9 | 9 | 0 | 1 | 3 | 102 |
| 8. Inclusive | 0 | 0 | 9 | 3 | 0 | 0 | 3 | x | 0 | 9 | 9 | 3 | 0 | 3 | 1 | 3 | 0 | 0 | 9 | 3 | 3 | 58 |
| 9. Autonomous | 9 | 0 | 0 | 3 | 0 | 9 | 9 | 0 | x | 0 | 1 | 0 | 9 | 0 | 0 | 0 | 0 | 0 | 3 | 3 | 1 | 47 |
| 10. Modular | 0 | 0 | 9 | 9 | 0 | 0 | 9 | 9 | 0 | x | 9 | 1 | 1 | 1 | 1 | 9 | 3 | 1 | 3 | 0 | 3 | 68 |
| 11. Spacious | 1 | 3 | 3 | 9 | 0 | 0 | 3 | 3 | 0 | 0 | x | 0 | 0 | 3 | 0 | 0 | 0 | 0 | 3 | 3 | 1 | 32 |
| 12. Comfortable | 1 | 1 | 1 | 3 | 9 | 0 | 9 | 3 | 0 | 9 | 9 | x | 1 | 9 | 9 | 9 | 3 | 3 | 3 | 3 | 3 | 88 |
| 13. Power source | 3 | 1 | 0 | 1 | 0 | 9 | 9 | 0 | 9 | 0 | 0 | 0 | x | 0 | 0 | 0 | 0 | 3 | 9 | 9 | 0 | 53 |
| 14. Waiting time | 9 | 0 | 0 | 1 | 0 | 0 | 0 | 0 | 3 | 0 | 3 | 0 | 3 | x | 0 | 0 | 0 | 0 | 9 | 3 | 1 | 32 |
| 15. Ergonomic | 0 | 0 | 0 | 3 | 0 | 0 | 3 | 3 | 0 | 9 | 3 | 1 | 0 | 0 | x | 3 | 1 | 3 | 0 | 0 | 1 | 30 |
| 16. Practical | 1 | 0 | 9 | 9 | 0 | 0 | 1 | 9 | 3 | 9 | 9 | 1 | 3 | 9 | 3 | x | 0 | 0 | 3 | 9 | 9 | 87 |
| 17. Appearance | 0 | 0 | 9 | 9 | 0 | 3 | 9 | 0 | 0 | 1 | 0 | 0 | 0 | 0 | 3 | 1 | x | 1 | 0 | 0 | 0 | 36 |
| 18. Resistant | 3 | 9 | 3 | 9 | 0 | 1 | 9 | 0 | 0 | 0 | 0 | 0 | 1 | 0 | 0 | 0 | 1 | x | 0 | 0 | 0 | 36 |
| 19. Available | 9 | 0 | 9 | 0 | 0 | 1 | 1 | 3 | 9 | 0 | 9 | 0 | 3 | 9 | 0 | 0 | 0 | 0 | x | 9 | 3 | 65 |
| 20. Widespread | 3 | 0 | 3 | 9 | 0 | 1 | 1 | 0 | 9 | 1 | 3 | 0 | 3 | 3 | 0 | 0 | 0 | 0 | 9 | x | 3 | 48 |
| 21. Can be integrated | 1 | 3 | 9 | 3 | 0 | 0 | 1 | 1 | 0 | 9 | 3 | 0 | 1 | 0 | 3 | 9 | 3 | 0 | 3 | 1 | x | 50 |
| INDEPENDENCE | 60 | 37 | 82 | 88 | 13 | 38 | 10 | 44 | 60 | 75 | 97 | 25 | 68 | 46 | 32 | 45 | 20 | 34 | 76 | 58 | 47 | |

**Table 5.** Microcar benchmark.

| | CITROËN AMI | RENAULT TWIZY E-TECH ELECTRIC | TAZZARI ZERO CITY | XEV YOYO | MICROLINO | INNOVATION |
|---|---|---|---|---|---|---|
| Length | 2410 mm | 2338 mm | 2795 mm | 2530 mm | 2519 mm | <2410 mm |
| Height | 1520 mm | 1 454 mm | 1450 mm | 1560 mm | 1501 mm | <1450 mm |
| Width | 1390 mm | 1381 mm | 1500 mm | 1500 mm | 1473 mm | <1381 mm |
| Weight | 490 kg | 446 kg | 650 kg | 850 kg | 513 kg | <446 kg |
| Battery | Lithium 5.5 Wh | Lithium 14 kWh | Lithium 14.2 kWh | Lithium 10.3 kWh | Lithium 14 kWh | >14.2 kWh |
| Engine | 6 kW | 13 kW | 15 kW | 7.5 kW | 12.5kW | >15 kW |
| Recharge | 3 h | 3.5 h | 3 h | 4 h | 4 h | <3h |
| Maximum speed | 45 km/h | 80 km/h | 90 km/h | 80 km/h | 90 km/h | ≤45 km/h |
| Battery life | 75 km | 100 km | 140 km | 150 km | 230 km | >230 km |
| Number of seats | 2 | 2 | 2 | 2 | 2 | 2 |
| Price | 5731 € | 12,000 € | 17,490 € | 14,900 € | 12,500 € | <5731 € |
| Doors | 2 | 2 | 3 | 3 | 2 | 3 |
| Trunk capacity | 0 L | 0 L | 445 L | 180 L | 230 L | >445 L |
| Supply voltage | 220 V | 220 V | 220 V | 220 V | 220 V | 220 V |
| Replaceable battery | No | No | No | Yes | No | Yes |
| Safety devices against accidents | Seat belts, ABS | Seat belts, airbag, lateral reinforcements | Seat belts, airbag, ABS | Seat belts, ABS, Airbag, lateral and frontal reinforcements | Seat belts | Seat belts, ABS, airbag, lateral and frontal reinforcements |

**Table 5.** *Cont.*

| | CITROËN AMI | RENAULT TWIZY E-TECH ELECTRIC | TAZZARI ZERO CITY | XEV YOYO | MICROLINO | INNOVATION |
|---|---|---|---|---|---|---|
| Energy recovery system | Yes | Yes | Yes | No | Yes | Yes |
| Steering diameter | 7.20 m | 6.8 m | 7 m | 8 m | 7.5 m | <6.8 m |
| Need driving license | ≥AM | ≥B1 | ≥B1 | ≥B1 | ≥B1 | ≥AM |
| TOP | 5 | 4 | 8 | 3 | 3 | |
| FLOP | 7 | 4 | 5 | 7 | 5 | |
| Δ | −2 | 0 | 3 | −4 | −2 | ≥4 |

**Table 6.** E–Bikes benchmark.

| | COWBOY 4 ST | VANMOOF X3 | TOWNIE 10D EQ | RALEIGH | CENTROS TOUR | COLEEN modern DB | ANGELL | INNOVATION |
|---|---|---|---|---|---|---|---|---|
| Weight | 19 kg | 20.8 kg | 25 kg | 30 kg | 26 kg | 24.5 kg | 15.9 kg | <15.9 kg |
| Battery | 35 V | 36 V | 36 V | 36 V | 36 V | 48 V | 36 V | >48 V |
| Replaceable battery | Yes | No | Yes | Yes | Yes | No | Yes | Yes |
| Brake | Hydraulic discs | Hydraulic discs | Hydraulic discs | Discs | Discs | Hydraulic discs | Discs | Hydraulic discs |
| Engine | 250 W | 250 W | 250 W | 250 W | 250 W | 750 W | 250 W | >750 W |
| Charge time | 3 h 20 min | 4 h | 4–5 h | 4–6 h | 4–5 h | 2 h 30 min | 2 h | <2 h |
| Maximum speed | 25 km/h | 25 km/h | 25 km/h | 20 km/h | 25 km/h | 45 km/h | 25 km/h | <25 km/h |
| Battery life | 70 km | 150 km | 100 km | 60 km | 206 km | 70 km | 90 km | >206 km |
| Rider height | 170–195 cm | 155–200 cm | 160–190 cm | 160–200 cm | 160–200 cm | 165–195 cm | 165–195 cm | ≥155–200 |
| Payload | 140 kg | 120 kg | 136 kg | 180 kg | 120 kg | 140 kg | 125 kg | > kg |
| Wheel type | Anti-drilling | Inner tube | Anti-drilling | Anti-drilling | Anti-drilling | Inner tube | Anti-drilling | Anti-drilling |
| Wheel | 27.5″ | 24″ | 27.5″ | 20″–21″ | 26 " | 27.5″ | 28″ | 24 < x > 25 |
| Chassis | Aluminium | Aluminium | Aluminium | Aluminium | Aluminium | Aluminium | Aluminium | Aluminium |
| Price | 2.490 € | 2.198 € | 3.799 € | 5.863 € | 3.195 € | 9.990 € | 2.860 € | <2.198 € |
| Luggage rack | Yes | Yes | Yes | Yes | Yes | Yes | Yes | Yes |
| Embedded gps | Yes | Yes | No | No | No | Yes | Yes | Yes |
| Bluetooth | Yes | Yes | Yes | No | No | Yes | Yes | Yes |
| Dedicated App | Yes | Yes | Yes | No | No | Yes | Yes | Yes |
| TOP | 7 | 8 | 6 | 3 | 4 | 6 | 8 | |
| FLOP | 3 | 4 | 2 | 8 | 6 | 4 | 3 | |
| Δ | 4 | 4 | 4 | −5 | −2 | 2 | 5 | >6 |

**Table 7.** Scooter benchmark.

| | Eswing ES | My Happy M2 | Ninebot KickScooter MAX G30E | Mi Electric Scooter Pro 2 | Urbetter S1 PRO | Ninebot KickScooter F40E | PRO-II EVO | INNOVATION |
|---|---|---|---|---|---|---|---|---|
| Length | 1045 mm | 1150 mm | 1167 mm | 1130 mm | 1050 mm | 1143 mm | 1195 mm | <1045 mm |
| Height | 1170 mm | 1160 mm | 1203 mm | 1180 mm | 1160 mm | 1160 mm | 1169 mm | <1160 mm |
| Width | 465 mm | 470 mm | 472 mm | 430 mm | 450 mm | 480 mm | 480 mm | <430 mm |
| Height when closed | 497 mm | 480 mm | 534 mm | 490 mm | 330 mm | 495 mm | 510 mm | <330 mm |
| Adjustable | Yes | No | No | No | Yes | No | No | Yes |
| Weight | 21.7 kg | 18 kg | 19.1 kg | 14.2 kg | 11 kg | 17.1 kg | 16.7 kg | <11 kg |
| Wheel | 8.5″ | 10″ | 10″ | 8.5″ | 8″ | 10″ | 10″ | >10″ |
| Wheel type | Tubeless | Tubeless | Tubeless | Honeycomb | Honeycomb | Tubeless | Tubeless | Honeycomb |
| Battery | Lithium 48V | Lithium 48V | Lithium 36 V | Lithium 36 V | Lithium 36 V | Lithium 36 V | Lithium 36 V | >48 V |
| Replaceable battery | No | No | No | No | No | No | No | Yes |
| Brake | Discs | Double disk | Electronic and drum | Electronic and disk | Discs | Electronic and disk | Electronic and disk | Electronic and disk |
| Engine | 450 W | 500 W | 350 W | 300 W | 350 W | 350 W | 350 W | ≥500 W |
| Charge time | 6 h | 4.5 h | 6 h | 8.5 h | 4 h | 6.5 h | 7 h | <4 h |
| Maximum speed | 35 km/h | 40 km/h | 25 km/h | 25 km/h | 30 km/h | 25 km/h | 25 km/h | <25 km/h |
| Battery life | 30 km | 50 km | 65 km | 45 km | 30 km | 40 km | 40 km | >65 km |
| Payload | 110 kg | 125 kg | 100 kg | 100 kg | 120 kg | 120 Kg | 100 kg | >125 kg |
| Price | 850 € | 785 € | 879 € | 550 € | 300 € | 600 € | 679 € | <300 € |
| Chassis | Aluminium | Aluminium | Steel and Carbon | Aluminium | Aluminium | Steel and Carbon | Magnesium alloy | Aluminium |
| Suspension | / | Front and rear | / | / | Front and rear | / | Front and rear | Front and rear |
| Bluetooth | No | Yes | Yes | Yes | No | Yes | Yes | Yes |
| Dedicated app | No | No | No | Yes | No | No | Yes | Yes |
| Maximum slope | 20° | 18° | 20° | 20° | 15° | 20° | 18° | ≥20° |
| Display | LED | LED | LED | LED | LCD | LED | LED | LED |
| Resistance to water | IPX5 | IPX4 | IPX5 | IP54 | IP54 | IPX5 | IPX4 | ≥IP54 |
| Cruise control | No | Yes | No | Yes | Yes | No | Yes | Yes |
| TOP | 6 | 10 | 6 | 11 | 11 | 7 | 8 | |
| FLOP | 8 | 5 | 11 | 6 | 8 | 7 | 5 | |
| Δ | −2 | 5 | −5 | 5 | 3 | 0 | 3 | ≥6 |

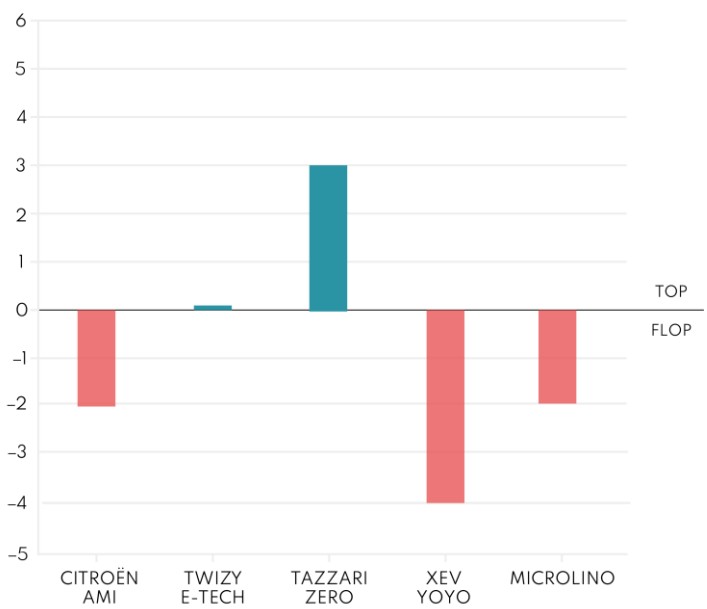

**Figure 10.** Microcar top-flop analysis.

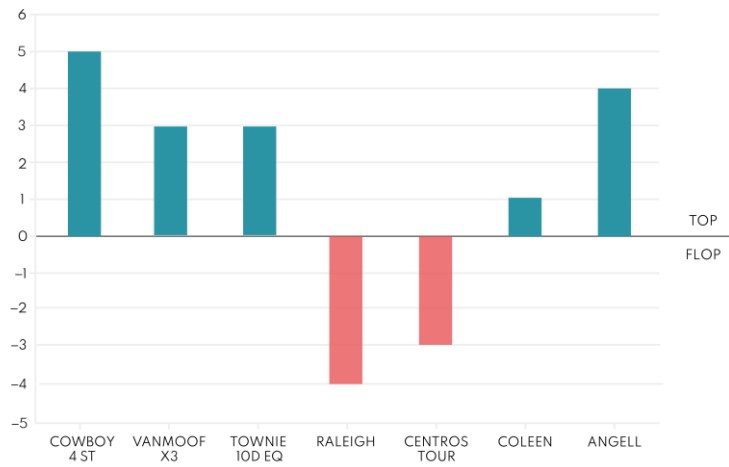

**Figure 11.** E-Bikes Top-Flop analysis.

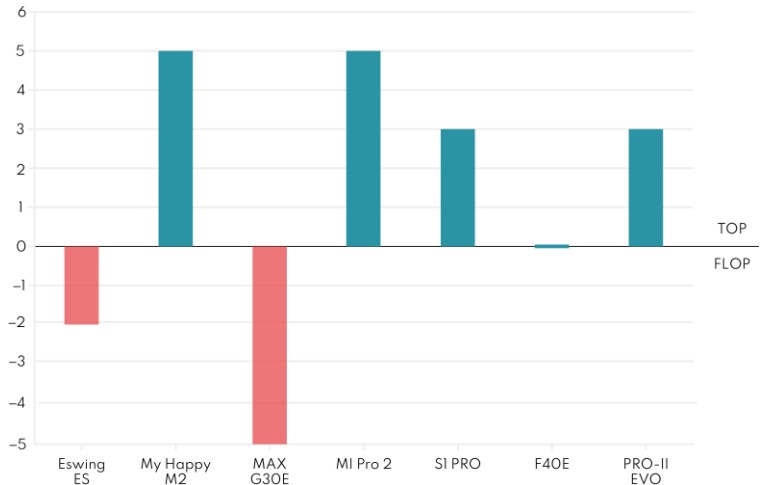

**Figure 12.** Scooter top-flop analysis.

After that, the technical characteristics obtained from the three benchmarking analyses were interpolated into the what-what matrix with the requirements derived from the QFD (as already described in Section 2.1.2). The goal of this matrix is to identify which design considerations should be prioritized (Table 8).

**Table 8.** What-how matrix.

|  | Dimensions | Battery Life | Weight | Number of Seats | Maximum Speed | Luggage Rack/Trunk | Charge Time | Replaceable Battery | Driving License Needed | Price |
|---|---|---|---|---|---|---|---|---|---|---|
| Economical | 6 | 4 | 2 | 2 | 2 | 0 | 6 | 4 | 0 | 10 |
| Autonomous | 6 | 10 | 6 | 0 | 8 | 0 | 0 | 6 | 0 | 4 |
| Waiting time | 8 | 6 | 0 | 4 | 8 | 0 | 10 | 10 | 0 | 0 |
| Available | 0 | 6 | 0 | 4 | 4 | 0 | 6 | 4 | 0 | 0 |
| Widespread | 10 | 8 | 0 | 0 | 0 | 0 | 6 | 0 | 2 | 0 |
| Practical | 10 | 8 | 8 | 6 | 4 | 8 | 10 | 10 | 6 | 0 |
| Inclusive | 6 | 0 | 0 | 2 | 0 | 6 | 0 | 0 | 10 | 6 |
| Compact | 10 | 6 | 0 | 8 | 2 | 6 | 6 | 4 | 0 | 4 |
| Spacious | 10 | 0 | 0 | 10 | 0 | 10 | 0 | 0 | 0 | 2 |
| Sustainable | 4 | 0 | 0 | 0 | 0 | 0 | 4 | 6 | 0 | 8 |
| Total | **70** | **48** | 16 | **36** | 28 | **30** | **48** | **44** | 18 | 34 |

*2.9. Product Architecture*

The needs that the project must have to provide innovation and new values to the cities of the future were inferred from the what-how and interrelationship matrices. The requirements that emerged led to the design of the project's technical and functional components, which were subsequently schematized into a generic layout (Figure 13).

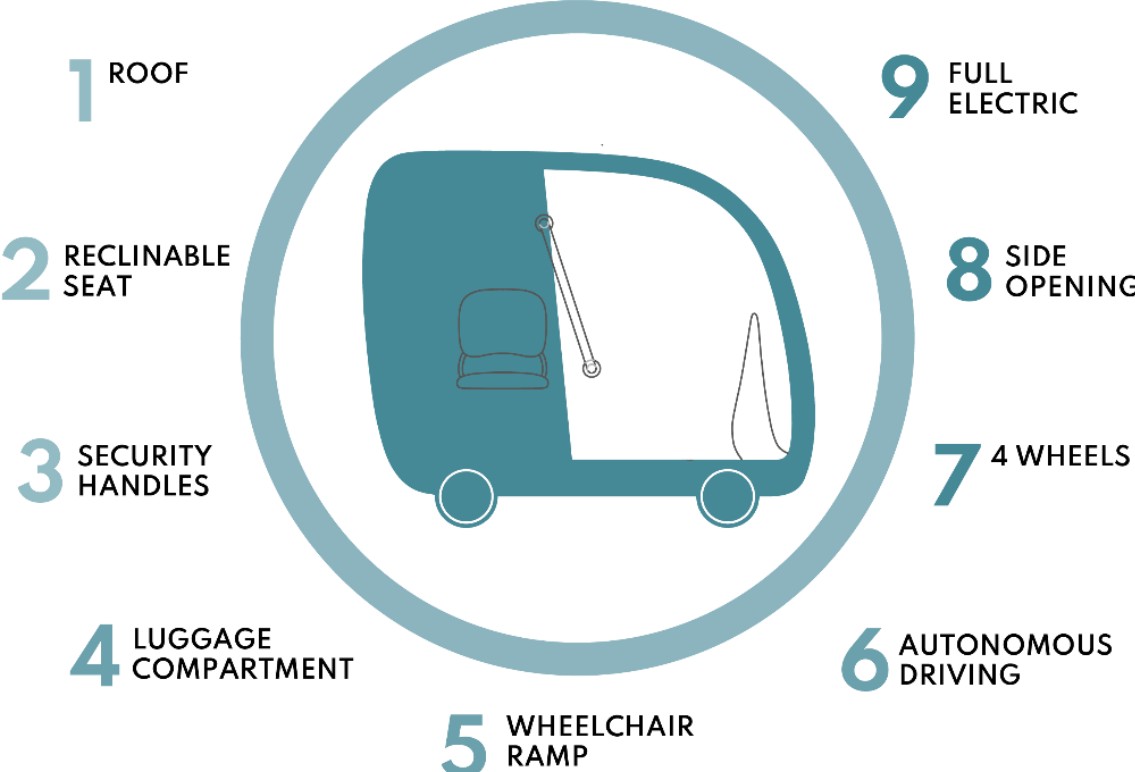

**Figure 13.** Product architecture layout.

Due to the growing sensitivity about cleanliness, the vehicle was designed to be covered and useable even in adverse weather, as well as have wide side openings for proper ventilation. The existence of bulkheads, which help protect against cold and poor weather, makes the opening optional.

Full electric mobility represents the mobility of the future and is especially suited to sharing, with the added benefit of reducing $CO_2$ emissions at the urban level. Due to the need to insert the wheelchair ramp in the centre (between the wheels) of the vehicle bed, engines were selected that could be built into the rear wheels to allow the bed itself to be lowered, but mostly to observe the norm on the maximum slope of the ramps for wheelchairs (8%), law 13/89, and decree 236/89. The batteries were arranged according to this logic; in fact, they are divided into two modules: one on the front of the frame and the other on the rear. The usage of 10 kWh lithium batteries allows for more calculations on the vehicle's theoretical range and viability, as it is an already well-implemented product as opposed to future generation batteries, which are currently being developed and tested.

Charging the vehicle is done by electromagnetic induction from a stationary vehicle. The principle of operation is energy exchange between two pads: one located on the ground and one in the underside of the vehicle [18,19]. SAE International (the automotive and aerospace industry's standardization authority) produced the SAE J2954 and SAE J2846/7 standards, which specify specifications for an energy exchange capable of recharging up to 11 kW with 94% efficiency and a pad-to-vehicle distance of up to 25 cm [20]. Induction charging is particularly suitable for self-driving vehicles because excellent alignment occurs between the two pads and thus high efficiency is achieved. This type of charging requires the implementation of power line and infrastructure, but this will necessarily have to be done given the increase of electric vehicles on the road in the near future [21]. Compared to vertical charging stations (columns), the selected solution requires less urban space clutter by being able to place the pads at strategic points in the city such as supermarkets, schools, airports and stations (Figure 14). This makes the best use of momentary stops (snack charging) by optimizing charging times.

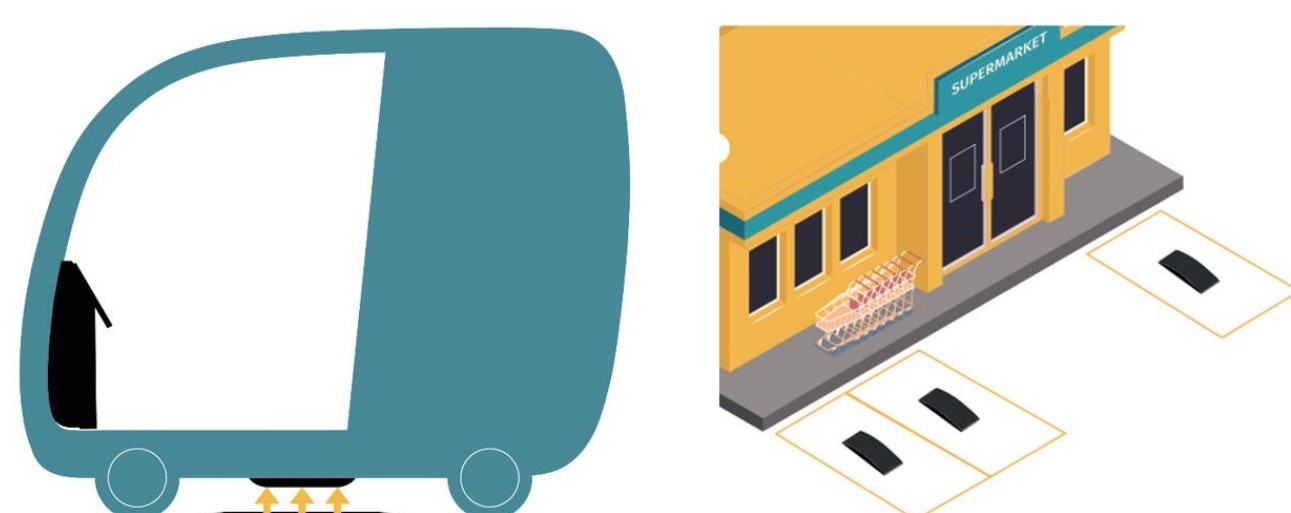

**Figure 14.** Charging concept.

The vehicle is designed to move in a future context, and in more connected and digitized cities. The smart city (Figure 15), in fact, is a driver of vehicle automation, improving road safety, traffic efficiency and energy consumption, while keeping transportation emissions more in check [22]. This is equipped with traffic lights interconnected to a Cloud system, digital signage, sensor technology, smart parking, cameras for obstacle recognition and smart monitoring to collect economic, environmental, safety and quality of life indicators and citizens' habits [23]. The combination of these advanced technologies will provide support to the vehicle (equipped with Lidar, radar, cameras, and ultrasonic

sensors), collecting specific data and facilitating communication between different vehicles, between vehicle users, and between road infrastructure [24,25].

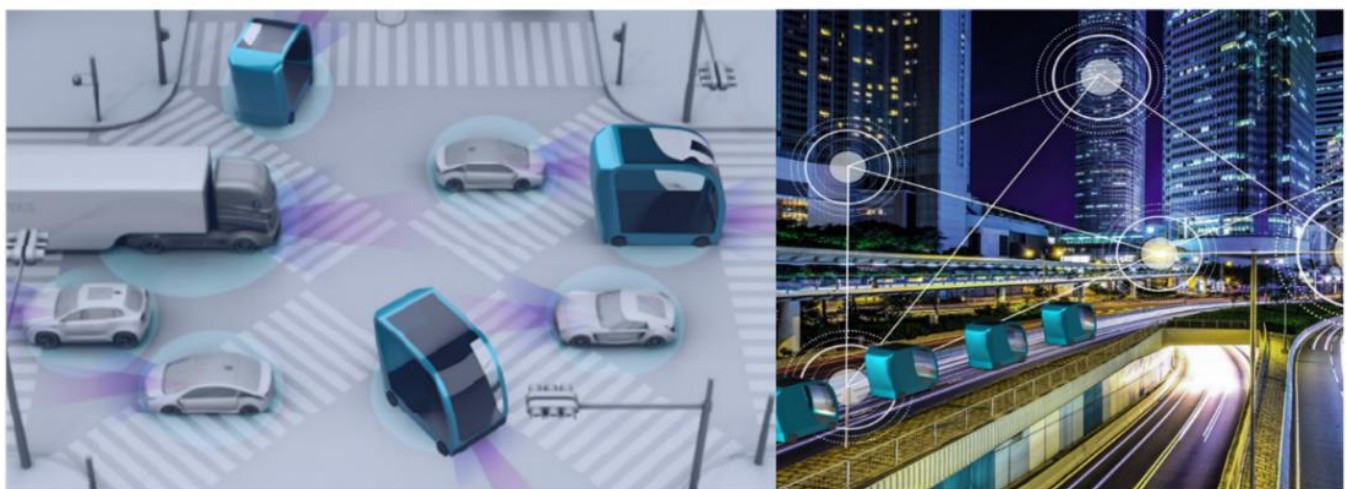

**Figure 15.** Smart city.

The overall dimensions (Figure 16) were studied using the general dimensions of the microcars as a reference and the exact measurements of each piece to be installed within the vehicle. The overall measurements (percentiles) of a person standing, wheelchairing, or sitting were validated, resulting in a vehicle that is internally compact but well-organized. To improve and maximise the livability of the area, the vehicle has a reclining seat that individuals who want to travel seated in comfort may utilise. When the seat is closed and fastened to the wall, wheelchairs may rest with the backrest and secure themselves with the belt beside the seat. Safety handles were included at many spots around the vehicle to allow for more dynamic use, including the tube on which the screen is fastened and the side of the reclining seat. This enables a risk-free vacation experience. Given the desired compactness, the car has an internal trunk consisting of three sloping shelves situated on the rear top, which hold luggage. Due to the size of the intended audience, the touch screen is tilt-adjustable, allowing it to be utilised standing up or straight from the wheelchair. The screen totally replaces the steering wheel; the address of the destination, which will be reached via level 5 autonomous driving [21,24], may be immediately typed into it. The knowledge of percentiles was useful for the demonstration of different possible configurations of use of the vehicle (Figure 17), to make it very versatile and tailored to the varied needs of citizens, and in particular by trying to facilitate people with reduced mobility, by simplifying the vehicle usage.

*2.10. Stylistic Design Engineering (SDE)*

Stylistic trends were examined, and sketches were created to describe the style idea. These are divided into four patterns (Figure 18): stone (angular lines), natural (more organic lines), retro (references to classic forms) and advanced (futuristic lines). The process of SDE is explained in Section 2.2. The styles are united by wide side openings, a front windshield that follows the body vertically, and the interiors described earlier in the product architecture section.

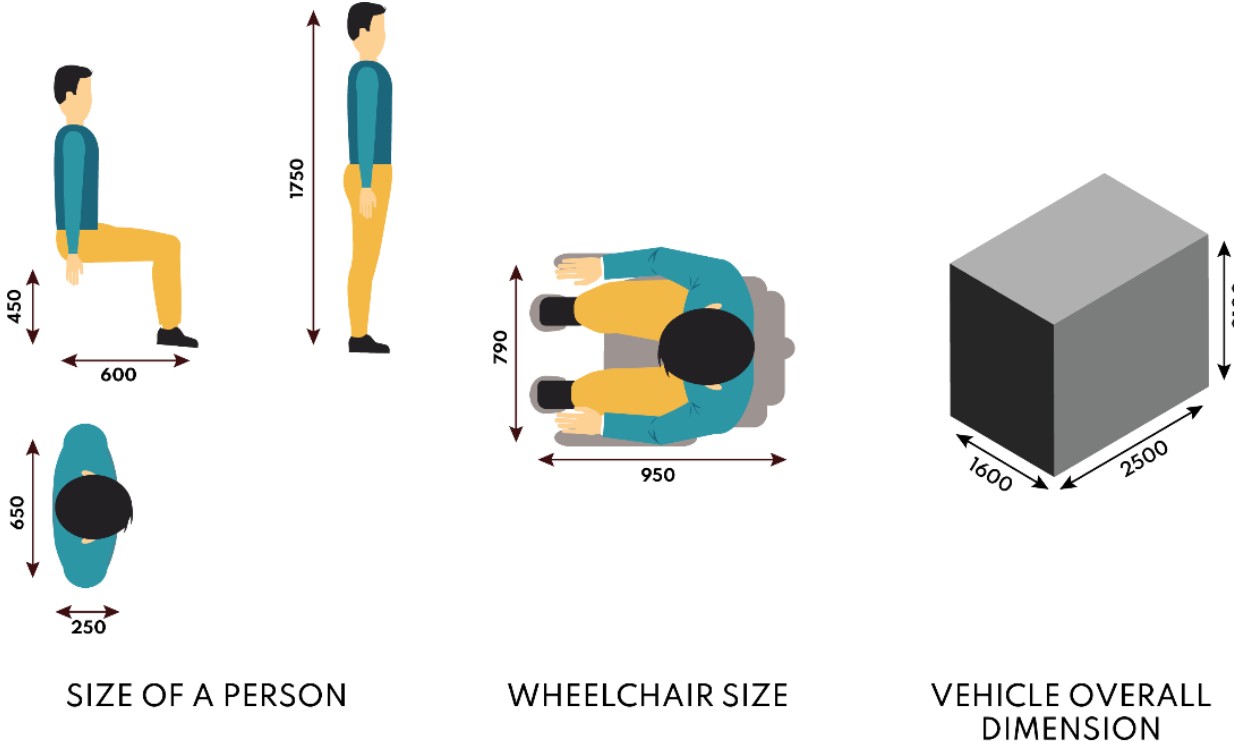

**Figure 16.** General measurements (measurements are expressed in millimeters).

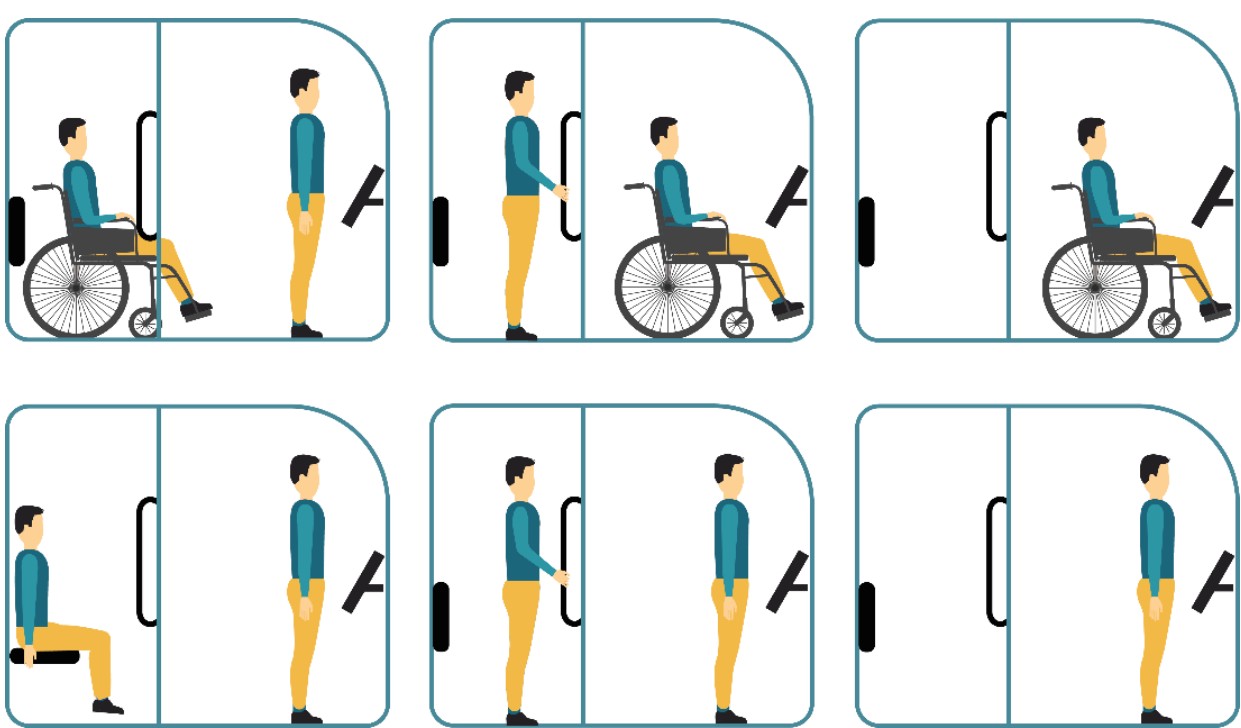

**Figure 17.** Configurations of use.

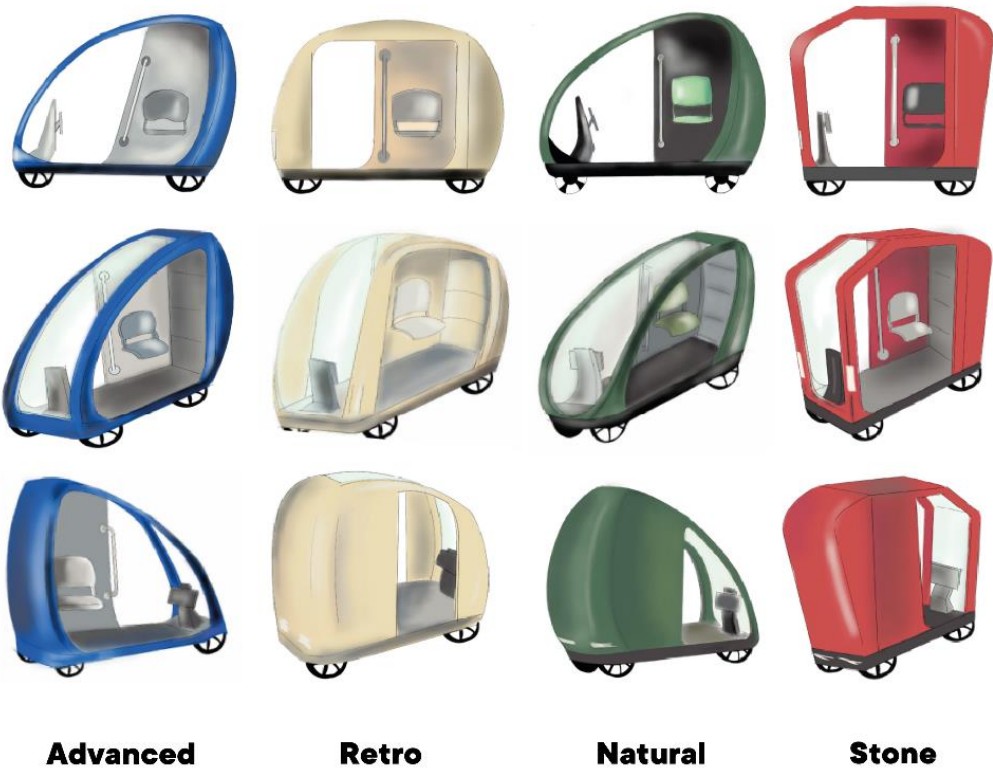

**Advanced**    **Retro**    **Natural**    **Stone**

**Figure 18.** SDE Drawings.

The choices that characterized the definition of the final proposal (Figure 19) are: maximum optimization of space and the right balance between organic and angular lines, maintaining for the interior layout, and what was previously defined in the product architecture. The sketch was dimensioned in the 2D table using software.

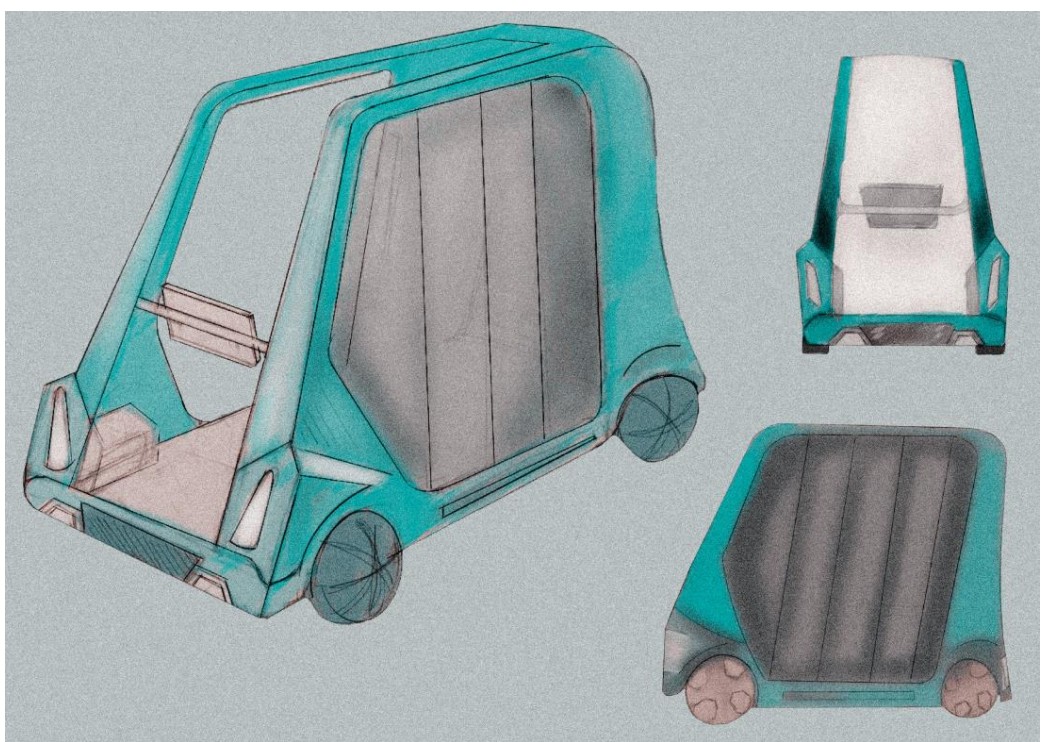

**Figure 19.** Final concept style.

## 3. Results

### 3.1. Design Outlines

The initial stage of project development is to revise and refine the sketches into precise 2D designs (Figure 20).

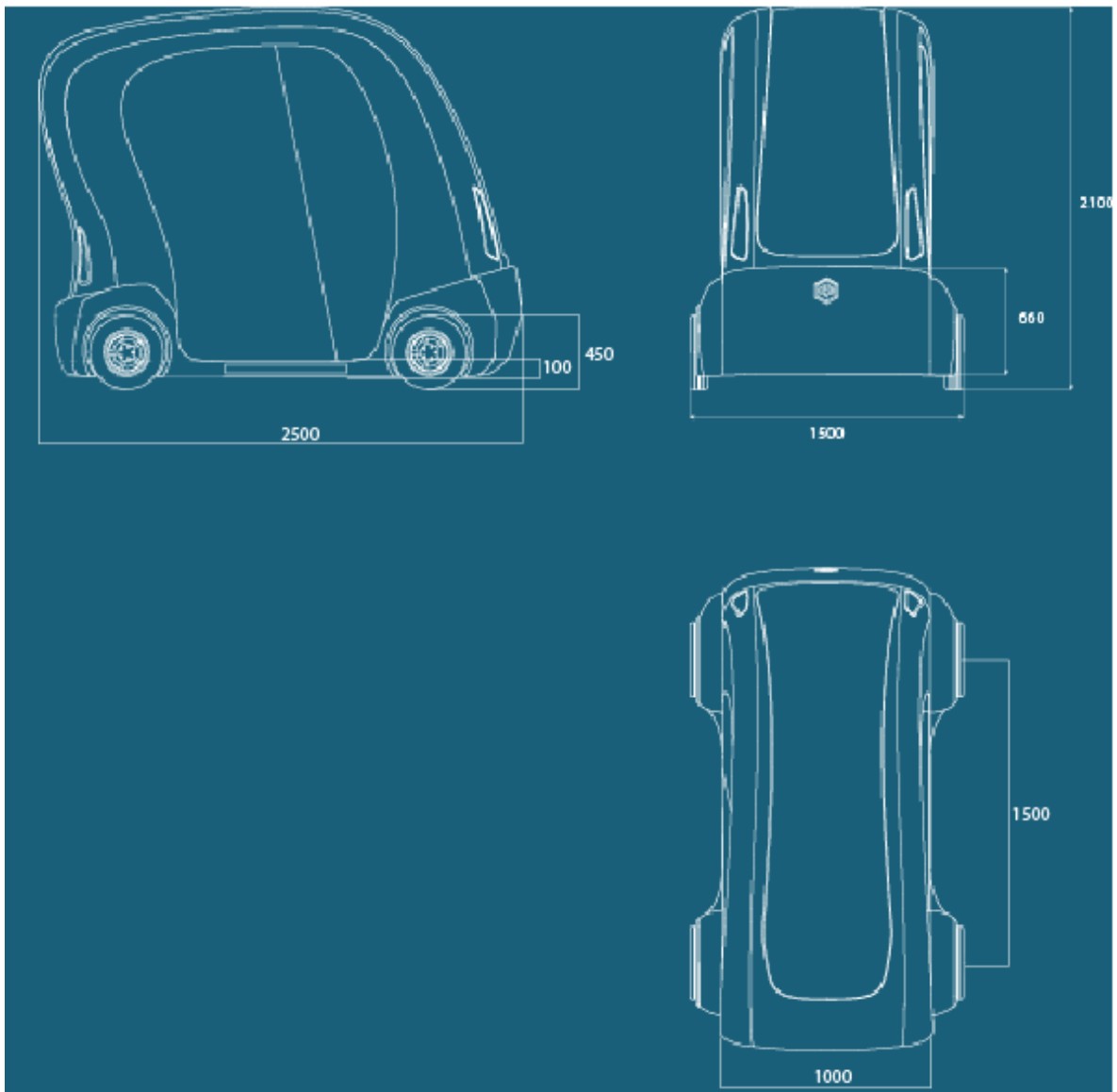

**Figure 20.** Final concept blueprints.

When compared to drawings, this method allows for a comprehensive picture of the design and makes it easier to identify any flaws. During this macro phase, a more engineering design approach was used, going into the vehicle's technological components.

The chassis was designed once the concept was defined and 2D comparison tables were created. Solid Edge, a CAD-type programme that enables for 2D and 3D design as well as finite element verification, was used to model the chassis. A rectangular hollow steel section from the UNI EN10219-2 family with dimensions (30 × 30 × 2) mm was chosen for the chassis. To fit the physique, it was kept basic. Figure 21 shows the analysis and subsequent schematization of the configuration of the battery packs in the front and back of the chassis, the rear motors integrated in the wheels, and the automated ramp positioned in the centre of the vehicle, as detailed in the Product Architecture section.

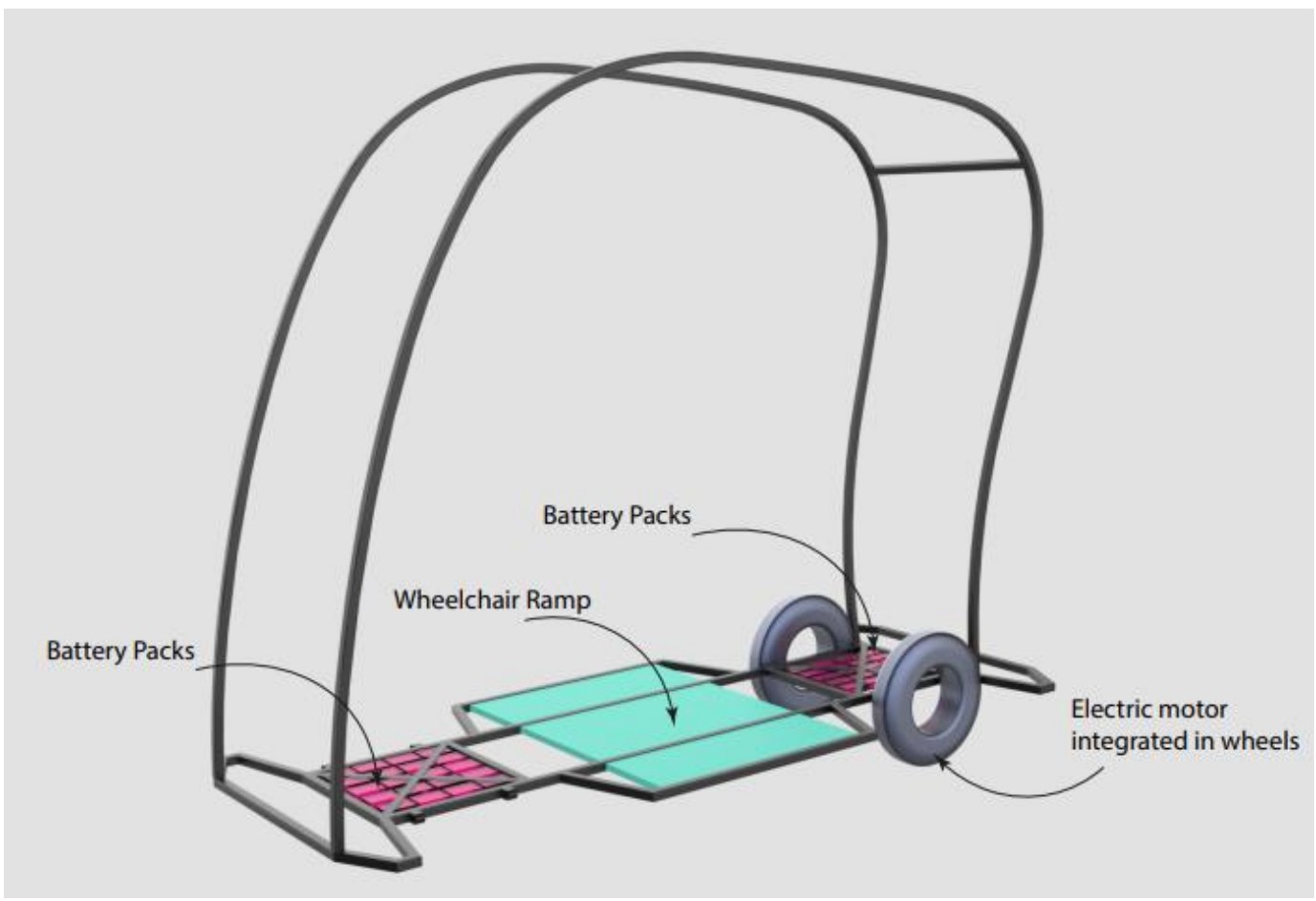

**Figure 21.** Chassis design.

To make sure the car was stable, the centre of gravity had to be calculated. The microcars that were previously evaluated during benchmarking were chosen, and the average weight was calculated to provide an approximation of the vehicle's weight (Table 9). The weight [26] was then broken up into the many components that make up the vehicle and finally, the centres of gravity for each component were calculated (Table 10). The entire centre of gravity (Figure 22) was then determined by considering three use cases: a single person, two people, and a person in a wheelchair with limited mobility (Table 11).

**Table 9.** Mass of vehicles examined in the benchmark and average mass.

|  | CITROËN AMI | RENAULT TWIZY E-TECH ELECTRIC | TAZZARI ZERO CITY | XEV YOYO | MICROLINO | AVERAGE MASS |
|---|---|---|---|---|---|---|
| Weight [kg] | 490 | 446 | 650 | 850 | 513 | 589.8 |

**Table 10.** Vehicle component mass and COG.

|  | **Mass [Kg]** | **Centre of Gravity [m]** |
|---|---|---|
| Frame | 138.6 | 0.15 |
| Propulsion | 198.8 | 0.18 |
| Electrical | 23.0 | 0.2 |
| Trim | 98.5 | 0.6 |
| Car body | 128.6 | 1.05 |
| Fluids | 2.9 | 0.18 |
| One person | 80 | 1 |
| Two people | 160 | 1 |
| Wheelchair user | 260 | 0.5 |

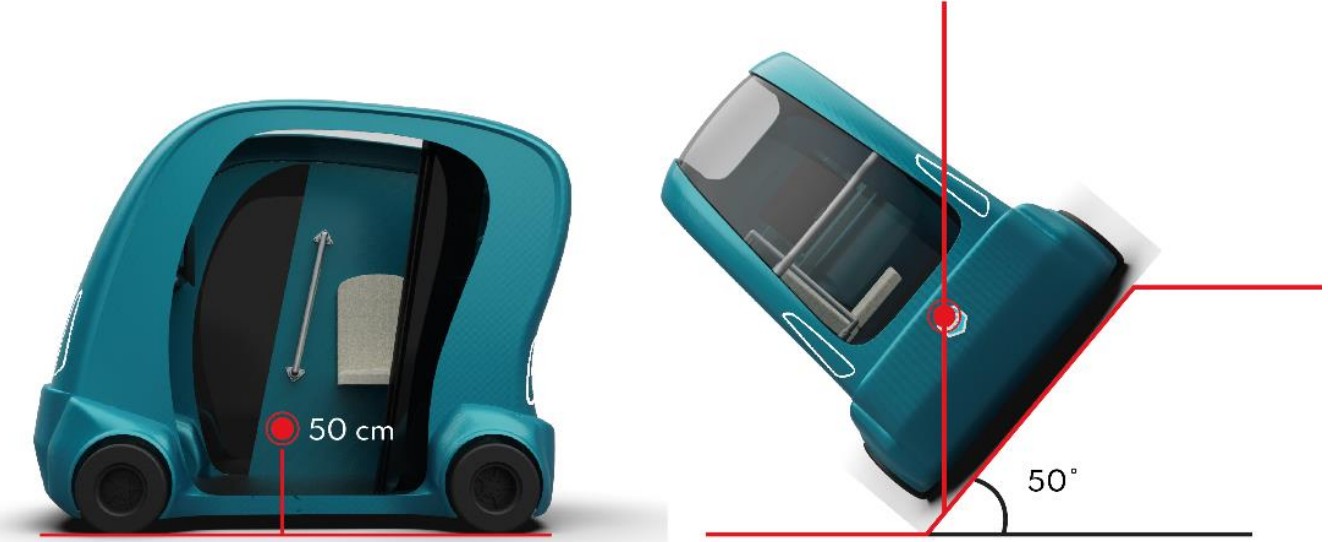

**Figure 22.** COG with one passenger on board.

**Table 11.** Maximum faceable side slope and total COG.

|  | **One Person** | **Two People** | **Wheelchair User** |
|---|---|---|---|
| Total mass of the vehicle [kg] | 590 | 590 | 590 |
| Wheel track [m] | 1.6 | 1.6 | 1.6 |
| Total centre of gravity [m] | 0.50 | 0.55 | 0.45 |
| Total mass [kg] | 670.4 | 750.4 | 850.4 |
| Maximum faceable side slope [°] | 58 | 55 | 60 |

The following formula is used to determine the height of the center of gravity (COG) (Figure 22):

Total height of COG [m] = (Σ mass of component * height of barycenter of the component)/total mass of the vehicle

In order to comply with the regulations on the maximum slope of wheelchair ramps (law 13/89 and decree 236/89), the height and length of the ramp (Figure 23) to be inserted into the vehicle was calculated to achieve an 8 % slope (maximum limit not to be exceeded).

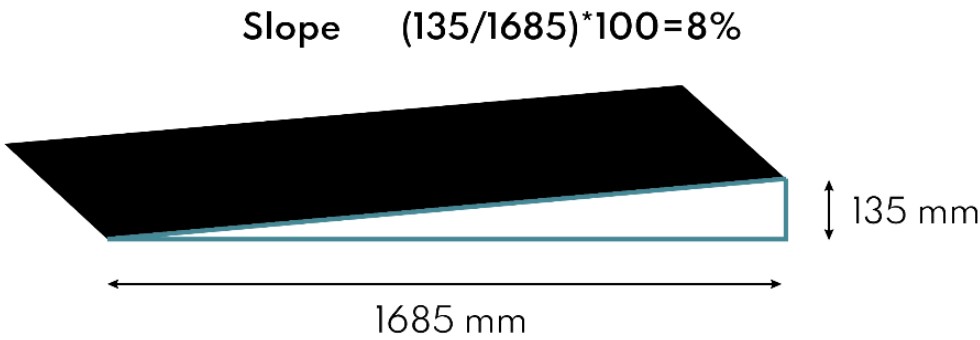

**Figure 23.** Wheelchair ramp specifications.

*3.2. Validation*

3.2.1. Testing

To examine the deformation of the frame under a particular load, several FEM-type calculations were performed using Solid Edge software. The material selected for the FEM investigation was structural steel. The first kind of study used a 1200 N load distributed on the centre section of the frame where the user would stand, resulting in a maximum displacement of $7 \times 10^{-5}$ mm, which was found in the central region (Figure 24). Fixed limitations were set at the frame's connection points to the remainder of the mechanical components. The maximum stress according to Von Mises was $1.9 \times 10^{-8}$ Mpa (Yield Stress: 262 MPa) based on the second study, which was performed under the identical constraint circumstances and loads (Figure 25). According to Von Mises, the maximum strain is $8.26 \times 10^{-14}$. Since both stresses were less than the prescribed safety limit, the frame design was deemed adequate.

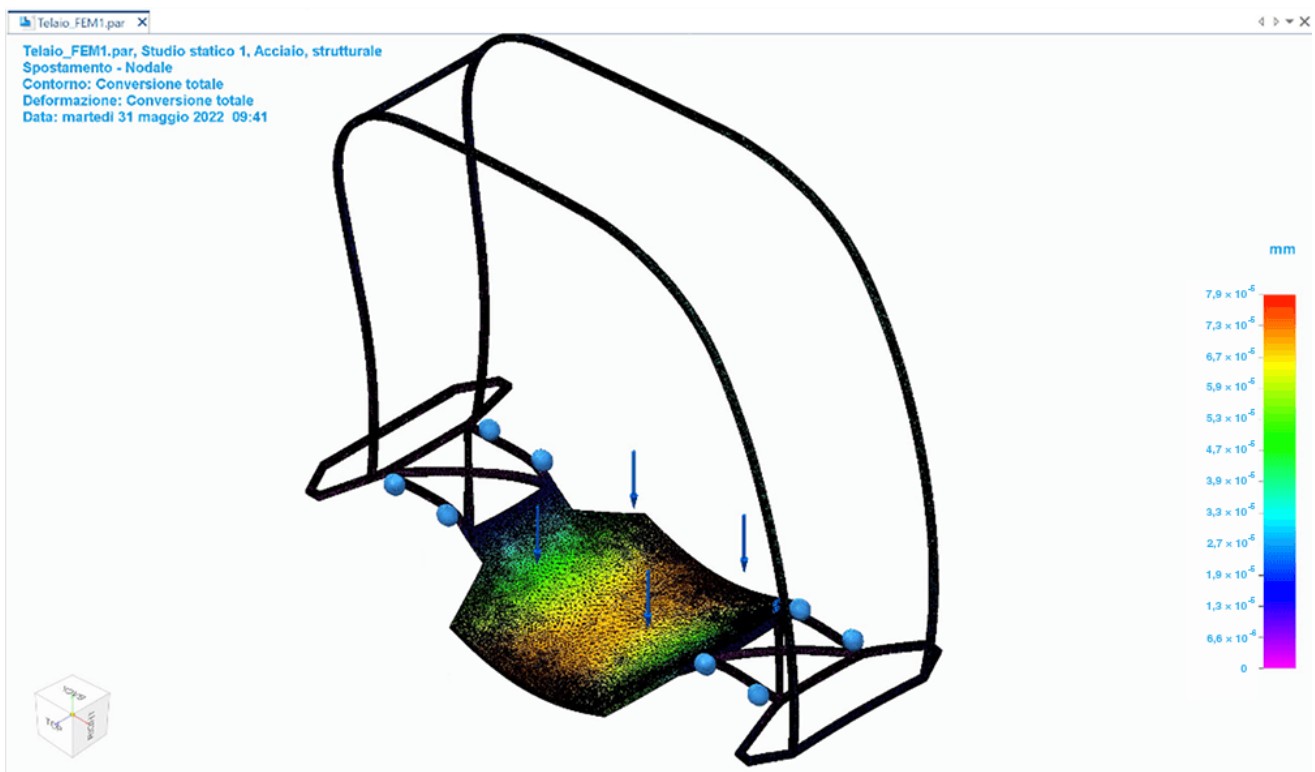

**Figure 24.** FEM displacement.

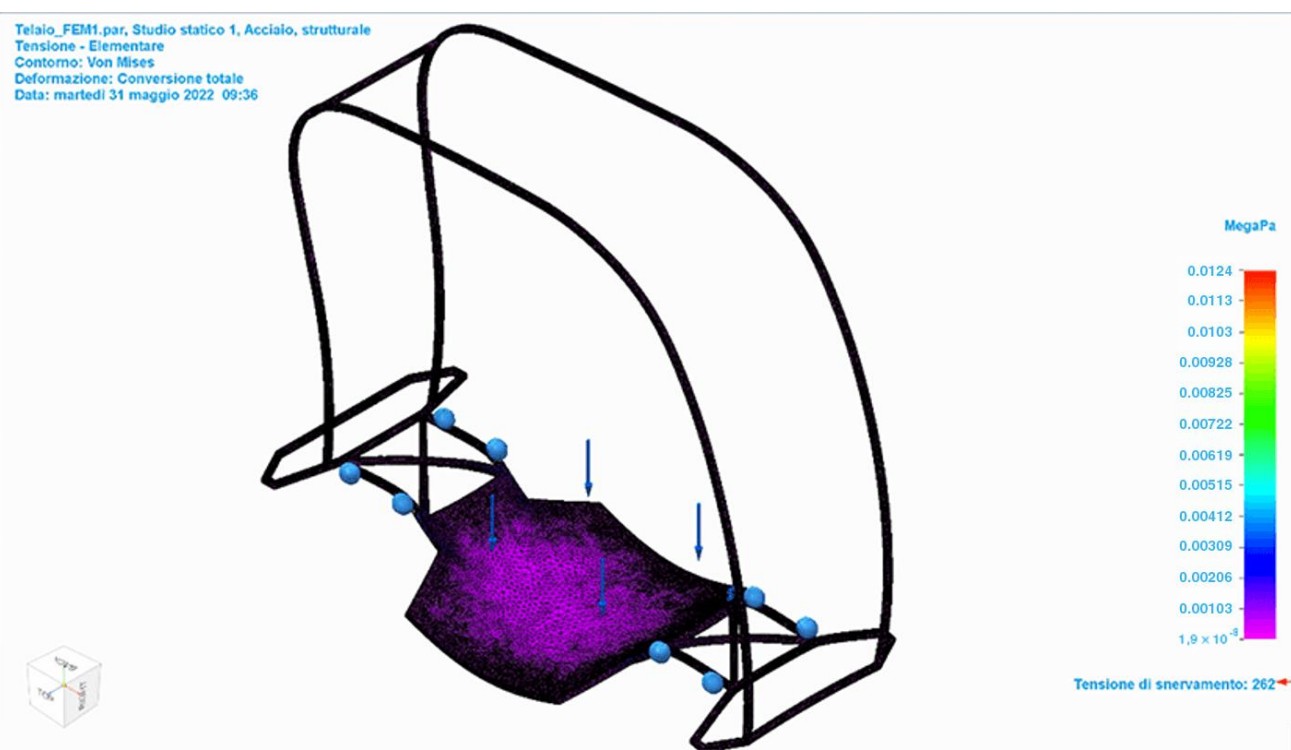

**Figure 25.** Von Mises maximum deformation.

### 3.2.2. Modeling

Once the shape of the chassis is defined, the 3D modeling phase of the vehicle can be started, using 3D Rhinoceros software (for 3D surface modeling made by Robert McNeel & Associates, a Seattle-based company in Washington state, USA). This phase takes a long time, as it is an important and fundamental step in the design of any product, but mainly because it requires advanced knowledge of the software being used. The goal to be achieved is to obtain a 3D model that is as faithful to reality as possible (Figure 26) and that is modeled perfectly according to the rules of 3D printing, to avoid interruptions during the latter.

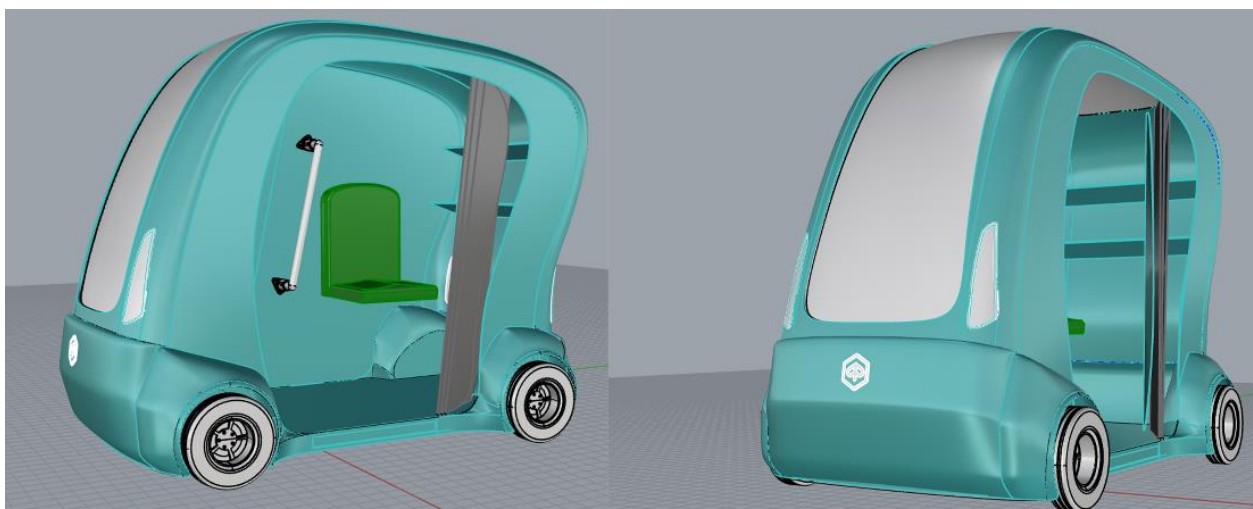

**Figure 26.** 3D modeling with Rhinoceros.

### 3.2.3. Rendering

Once the 3D model of the vehicle is obtained, the rendering phase begins, using Keyshot software. To achieve an optimal aesthetic impact, final materials and possibly textures are applied to the various parts of the model to add details to the scene. In Figure 27 different viewing perspectives of the vehicle have been rendered, and some of the interior elements can be seen, including the reclining seat and the sloping shelves for objects. Moreover, the interaction with the user and how they can access the vehicle through the mechanized ramp is clearly visible (Figure 27).

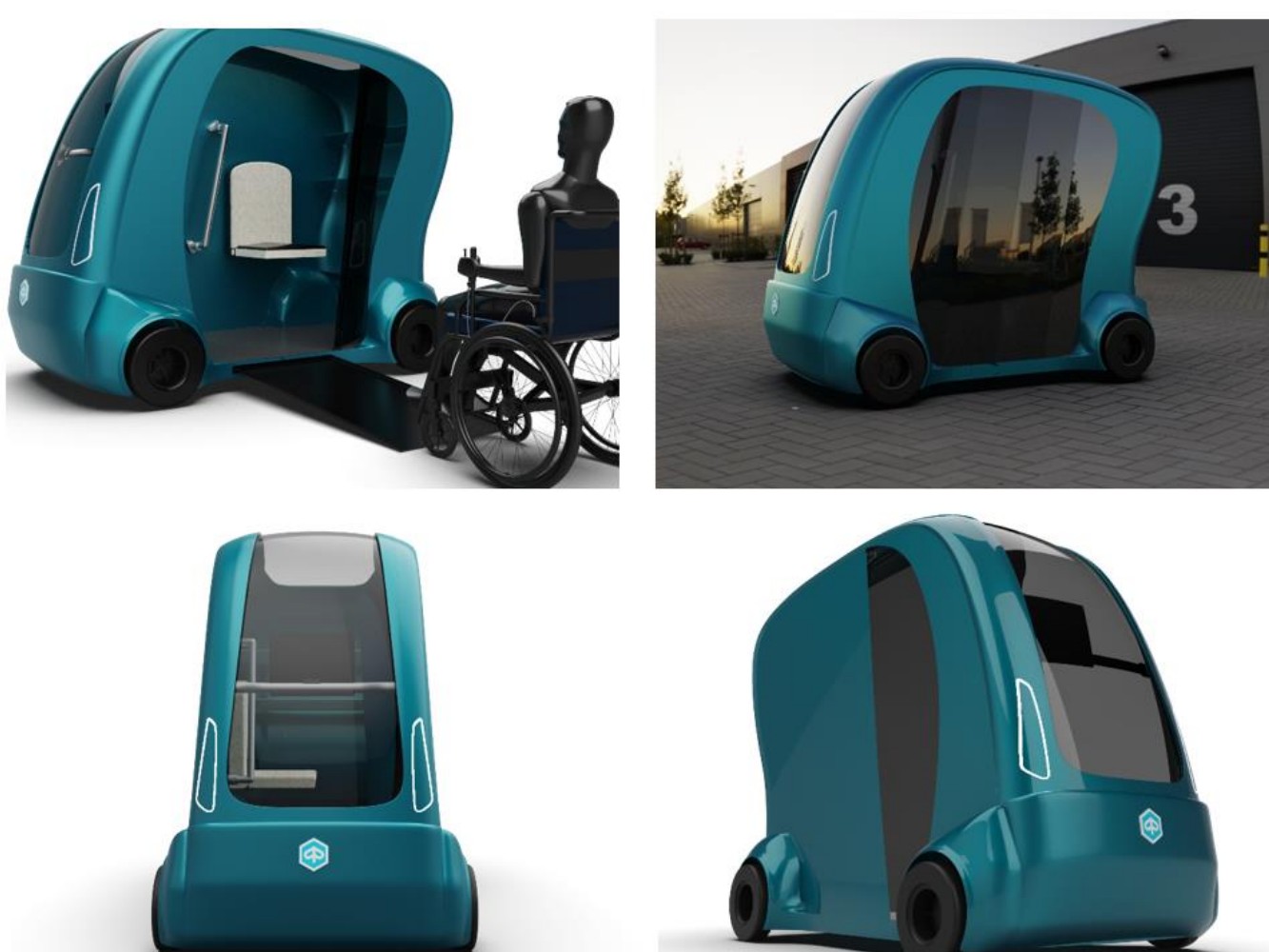

**Figure 27.** 3D rendering of the final concept.

### 3.2.4. Prototyping

The Ender 3 Printer (Fused Filament Fabrication technology) and the CURA slicing software were used to 3D print the scale model in white PLA. The model (Figure 28) was printed in multiple components that were then glued together. The pieces were post-processed (surfaces were sanded with sandpaper) to eliminate the "stepped" effect, which is common in additive printing.

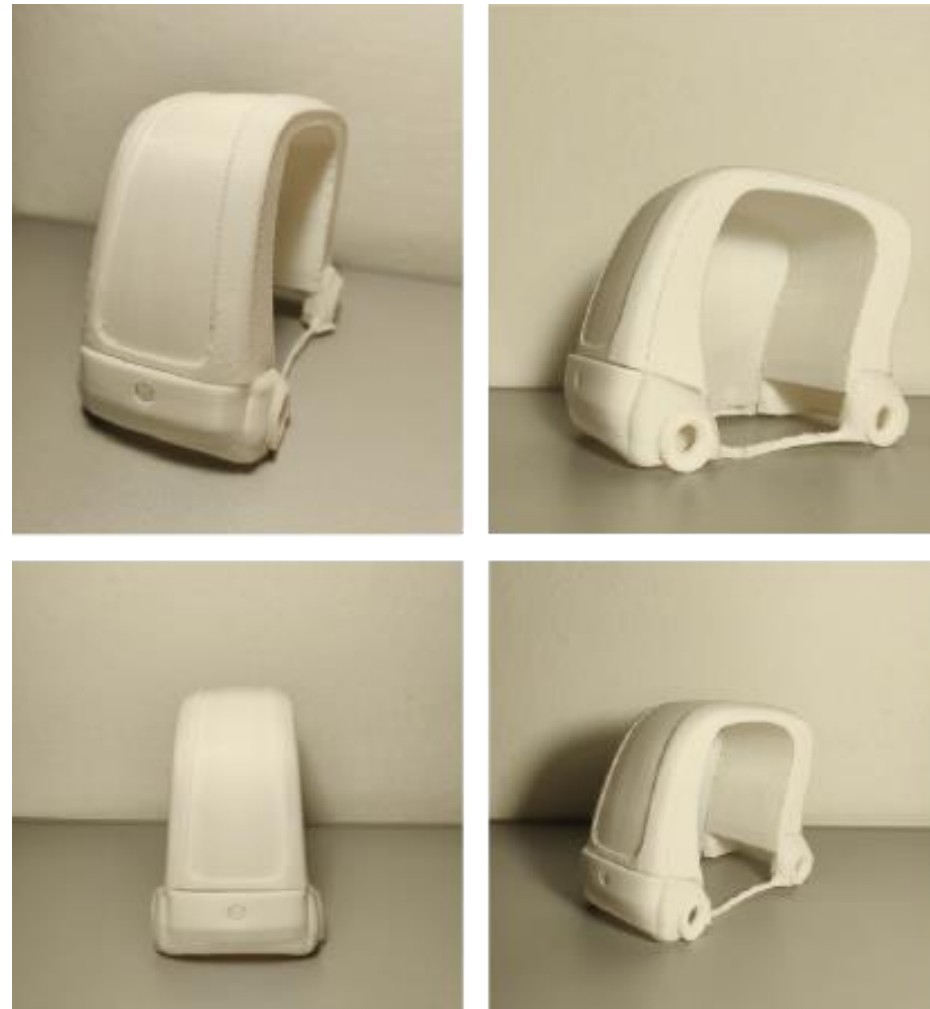

**Figure 28.** PLA 3D printed prototype of the final concept.

## 4. Discussion

*Proposed Improvements*

The effectiveness of the IDeS methodology was demonstrated in the development and application of the case study presented, using development tools and including management, technical and aesthetic phases (through QFD and SDE). It is the most comprehensive way to achieve and optimize an entire industrial project, limiting further changes before the start of production and satisfying the target market as much as possible.

Through the IDeS method, the goal of designing an innovative vehicle that does not yet exist in the market was achieved.

One of the project's primary criteria is level 5 autonomous driving, a technical and digital advancement that can accommodate the most vulnerable societal groups' demands for mobility. It is anticipated that by 2060, technology will advance sufficiently to enable widespread, as opposed to niche, urban use [27]. It is necessary to adapt the way we think about mobility in order to accommodate those with restricted mobility, broadening our perspective and aiming for more accessible, inclusive, and open communities for all residents. Data privacy will need to be ensured by the cybersecurity required to allow this citywide transition [22]. Self-driving vehicles and connected, smart streets offer a possibility to steer the evolution of transportation in the direction of a global society that is more sustainable [3]. The vehicle's design aims to open new urban daily living scenarios, built on the concepts of sustainability and freedom, and removing not only physical but also psychological boundaries.

## 5. Conclusions

The following technical criteria came from the what-how matrix as essential for vehicle design: size, battery life, number of seats, charge time, and trunk. The primary needs of the new product are autonomy, inclusivity, and sustainability, which makes the vehicle fully innovative. The car is environmentally friendly since it has an all-electric power supply and can move and recharge independently via induction within smart cities. The proportions of the vehicle were determined by studying the percentiles of persons with limited mobility who use wheelchairs, accessibility criteria via a ramp, and attempting to assure safety and stability. The concept of sustainability in modern society goes beyond environmental sustainability and aims to achieve a level of social inclusion for which there is a growing need and necessity.

**Author Contributions:** L.F. reviewed the group work along all stages of development and gave overall approval. G.G. was in charge of general project reviews and management of the paper's implementation. M.A., S.F. and E.R. handled the development of the project from the initial stages throughout its duration. All authors have read and agreed to the published version of the manuscript.

**Funding:** This research received no external funding.

**Data Availability Statement:** Not applicable.

**Acknowledgments:** The materials and machines used for the developing of the prototypal phase was granted by the DIN—Department of Industrial Engineering at Alma Mater Studiorum Università di Bologna.

**Conflicts of Interest:** The authors declare no conflict of interest.

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
