# Peer review of "Design of an Autonomous, Sustainable Sharing Mobility Solution Aimed to Mobility-Disabled Individuals"

_inventions, doi:10.3390/inventions8010044_

Round 1
Reviewer 1 Report
The manuscript entitled “Design of an Autonomous, Sustainable Sharing Mobility Solution aimed to Mobility-Disabled Individuals” is well written and logically structured. The presented contents are quite interesting and within the scope of inventions. However, there are too many figures in this manuscript, and some of them can be converted to tables. The reviewer recommends the minor revision before publication. The comments are:
1. Figures 6, 9, 14, 16, 18, 20, 30, 31, and 32 can be converted to tables.
2. Please remove the interface of software from Figure 37. The authors can simply show the 3D modelling.
3. English is acceptable but can be further polished.
4. Affiliations of the authors should be added into the revised manuscript.
Reviewer 2 Report
Dear Authors,
The article submitted for review is a very interesting study presenting the process of developing a special micromobility vehicle design involving an innovative fusion of design approaches. The thoroughness of the description and the detail of the data and considerations shown make this article very valuable and interesting for different groups of readers. I have only a few comments relating to editing issues, which I list in points:
1. For full readability, it is worth writing explicitely at the end of the Introduction what is the main purpose of the article.
2. various statistics are cited in the article and mobility is described in quantitative and qualitative terms - due to the international nature of the journal, it should either be written in the introduction that all these data refer to Italy or mentioned each time, as far as the text is concerned. On the other hand, in the figures - in their captions - due to the postulate of their self-explanatory nature, this should be indicated each time (this refers to figures 8, 10, 11).
3. check the uniformity of the style of references to the literature (e.g. line 168).
4. from line 270 onwards there are wrong references to figures - the reference to figure 8 is specified as to figure 6, etc.
5. several figures show the advantages and disadvantages of different solutions (e.g. figures 7 and 9). It is not clear whether this was developed by the team or taken from the literature (which literature?) the same applies to benchmarking (Figs 12 and 13). Sources should either be added or clarified in the text.
Many congratulations on a very interesting article.
